# RidgeLoRA: Matrix Ridge Enhanced Low-Rank Adaptation of Large Language Models

**Junda Zhu**[1]    **Jun Ai**[1]    **Yujun Li**[2*]    **Yichun Yin**[2]
**Yasheng Wang**[2]    **Lifeng Shang**[2]    **Qun Liu**[2]
[1]Beihang University    [2]Huawei Noah's Ark Lab
junda_zhu@outlook.com    liyujun145@gmail.com

https://github.com/chuhac/RidgeLoRA

## Abstract

As one of the state-of-the-art parameter-efficient fine-tuning (PEFT) methods, Low-Rank Adaptation (LoRA) enables model optimization with reduced computational cost through trainable low-rank matrix. However, the low-rank nature makes it prone to produce a decrease in the representation ability, leading to suboptimal performance. In order to break this limitation, we propose RidgeLoRA, a lightweight architecture like LoRA that incorporates novel architecture and matrix ridge enhanced full-rank approximation, to match the performance of full-rank training, while eliminating the need for high memory and a large number of parameters to restore the rank of matrices. We provide a rigorous mathematical derivation to prove that RidgeLoRA has a better upper bound on the representations than vanilla LoRA. Furthermore, extensive experiments across multiple domains demonstrate that RidgeLoRA achieves better performance than other LoRA variants, and can even match or surpass full-rank training.

## 1 Introduction

Large Language Models (LLMs) with large number of parameters [1–8] have demonstrated exceptional performance in natural language generation tasks. These models acquire their primary knowledge during the pre-training phase, through training on massive high-quality datasets from both real-world corpora or model-generated synthetic data. To align with real-world scenarios [9], LLMs also require supervised fine-tuning (SFT). Traditionally, this fine-tuning process employs full-parameter (also full-rank) training (FFT) to achieve optimal performance. However, this approach demands substantial computational resources.

When adapting LLMs for downstream tasks, training is often constrained by limited computational resources, calling for efficient and lightweight solutions. Parameter-Efficient Fine-Tuning (10, 11, PEFT) methods achieve comparable performance to full-parameter fine-tuning with minimal cost. Low-rank adaptation (LoRA, 12), as a representative PEFT method, is widely adopted due to the fact that downstream tasks largely rely on the generic capabilities developed in pre-training. To maintain model performance, LoRA's initial state should also align with the original model's output, which we refer to as *"the transform-calibrating restriction"*.

However, vanilla LoRA, while significantly reducing the number of trainable parameters, is often criticized for its lower performance ceiling [13–15]. Its low-rank nature results in significantly lower representation capability compared to full-parameter fine-tuning, making it prone to underfitting in downstream tasks [14]. Existing LoRA variants [16–21] primarily focus on matrix decomposition or

---

* Corresponding author.

39th Conference on Neural Information Processing Systems (NeurIPS 2025).

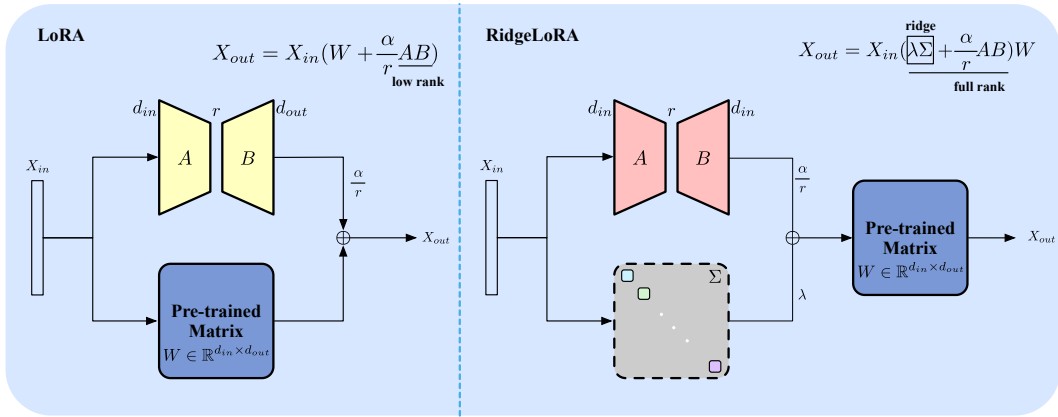

Figure 1: Differences between LoRA (left) and RidgeLoRA (right): As is depicted, though LoRA reduces the number of parameters to be trained, the matrix rank severely shrinks. In order to be comparable with full rank training, the proposed RidgeLoRA introduces a matrix ridge (formalized as $\lambda\Sigma$) to complement the rank of the trainable parameters.

numerical stability, proposing better parameter initialization methods or updating strategies. However, these methods lack theoretical investigation on full-rank matrix approximation and the fundamental challenges of representation ability in low-rank settings. Additionally, exploring architectural alternatives beyond the vanilla LoRA framework could potentially unlock new opportunities for improvement.

Unlike existing works, RidgeLoRA incorporates a full-rank module to achieve performance comparable to full-parameter training. We redesign the architecture by replacing the parallel connection in vanilla LoRA with a series connection module. Inspired by the Matrix Ridge algorithm [22], we enhance vanilla LoRA by incorporating a ridge term, thus introducing RidgeLoRA. The architectural differences between RidgeLoRA and vanilla LoRA are illustrated in Figure 1. Based on observations from previous works [17, 18] and our experiments, appropriate initialization methods can significantly improve low-rank training performance. As RidgeLoRA adopts a series connection architecture, it enables different parameter initialization approaches even under *"the transform-calibrating restriction"*. RidgeLoRA achieves comparable performance to full-parameter training at minimal cost without introducing additional computation and parameters. Our main contributions are summarized as three-fold:

- We propose RidgeLoRA, a novel LoRA variant that replaces parallel connection with series connection, enabling better parameter initialization and enhanced representation capability.
- We introduce a diagonal ridge term alongside the low-rank matrices, inspired by the matrix ridge algorithm [22], which improves approximation flexibility while maintaining computational efficiency, to better represent the high-rank updates of LLM training.
- We provide theoretical analysis of RidgeLoRA's representation capability and demonstrate its superior performance through extensive experiments across multiple datasets.

## 2 Related Works

### 2.1 Matrix Low-Rank Decomposition and LLMs

Matrix decomposition aims at optimizing the following objective:

$$\min \|W - W_r\|_F^2,$$

where $W \in \mathbb{R}^{d_{in} \times d_{out}}$ is the target matrix, $W_r$ represents its low-rank approximation with rank $r < \min(d_{in}, d_{out})$ and $\| \cdot \|_F$ denotes the Frobenius norm. Given a restricted rank $r$, Singular value decomposition (SVD, 23) has a bounded (also the minimum) error, which justifies its wide application, whose details can be found in Appendix A.3. In order to accelerate the inference

| LoRA Method | Number of Parameters | Weight Initialization Complexity | Forward Formalization | Computation Complexity |
|---|---|---|---|---|
| *FFT* | $d_{in} \times d_{out}$ | $\mathcal{O}(1)$ | $X_{in}W'$ | $\mathcal{O}(d_{in}^2 d_{out})$ |
| *LoRA* [12] | $r \times (d_{in} + d_{out})$ | $\mathcal{O}(rd_{in})$ | $X_{in}(W + \frac{\alpha}{r}AB)$ | $\mathcal{O}(d_{in}^2 d_{out})$ |
| *DoRA* [16] | $r \times (d_{in} + d_{out}) + d_{out}$ | $\mathcal{O}(rd_{in}d_{out})$ | $(\frac{\|W\|_2}{\|W + \frac{\alpha}{r}AB\|_2} - 1)XW + \frac{\|W\|_2}{\|W + \frac{\alpha}{r}AB\|_2} \cdot XAB \cdot \frac{\alpha}{r}$ | $\mathcal{O}(d_{in}^2 d_{out})$ |
| *PiSSA* [17] | $r \times (d_{in} + d_{out})$ | $\mathcal{O}[\max(d_{in}, d_{out}) \cdot \min(d_{in}^2, d_{out}^2)]$ | $X_{in}(W_{res}^{\mathbf{P}} + \frac{\alpha}{r}AB)$ | $\mathcal{O}(d_{in}^2 d_{out})$ |
| *KaSA* [18] | $r \times (d_{in} + d_{out} + 1)$ | $\mathcal{O}[\max(d_{in}, d_{out}) \cdot \min(d_{in}^2, d_{out}^2)]$ | $X_{in}(W_{res}^{\mathbf{K}} + \frac{\alpha}{r}A\Sigma_r B)$ | $\mathcal{O}(d_{in}^2 d_{out})$ |
| *RidgeLoRA* | $r \times 2d_{in} + (d_{in} + 1)$ | $\mathcal{O}(r^2 d_{in})$ | $X_{in}(\lambda\Sigma + \frac{\alpha}{r}AB)W$ | $\mathcal{O}(d_{in}^2 d_{out})$ |

Table 1: Comparisons between FFT and other LoRA methods, here we focus on analyzing the number of parameters from the perspective of a single matrix. The comparison here also considers the complexity of weight initialization, where some methods may conduct SVD or matrix multiplications. $W_{res}^{\mathbf{P}}$ and $W_{res}^{\mathbf{K}}$ denote different ways to initialize $W_{res}$ in their works. RidgeLoRA is showcased with simple initialization method, small number of parameters, and will not increase the computation and memory requirements during training.

speed, SVD-LLM [24] utilizes SVD compression method together with data whitening [25] to minimize $\|XW - XW_r\|_F^2$ to compress LLMs. MoDeGPT [26] conducts detailed analysis of each matrix calculation operation inside an LLM and selects the corresponding decomposition method accordingly, namely SVD, Nyström approximation [27] and CR decomposition [28]. As an inspiration of this paper, Matrix Ridge [22] proposes an algorithm that approximates a positive semi-definite matrix using a combination of an incomplete matrix decomposition and a ridge term. This achieves tighter approximation than both incomplete Cholesky decomposition [29] and incomplete spectral decomposition, while ensuring that the condition number of the approximated matrix does not exceed that of the original matrix.

## 2.2 Parameter-Efficient Fine-Tuning

**LoRA and its Variants**    LoRA and its variants [12, 16–21, 30–35] train extra low-rank weights on top of the model, which makes low-rank matrix decomposition naturally suitable for the improvement of it. Considering *"the transform-calibrating restriction"*, advanced LoRA variants like PiSSA [17], LoRA-GA [30], MiLoRA[31], LoRA-XS [32] and KaSA [18] conduct SVD on the original matrices to endow the low-rank matrices with better initializations, thus achieving better performance. Furthermore, some works focus more on the numerical stability of matrices, rsLoRA [34] and proposes a better setup of the scaling factor from a statistical point of view. To achieve training stability, DoRA [16] decomposes the updates of matrices to magnitude factor and direction factor, hereby adding a scaling factor during training. VeRA [35] adds learnable scaling vectors which can be updated and tune frozen random matrices across layers, which changes a bit of the architecture.

Apart from parameter-based methods, LoRA+ [20] assigns the matrix $B$ initialized with all zeros with larger learning rate, LoRA-Pro [19] updates the matrices from the perspective of transformation invariance. These works all focus on how to deal with the gradients in the optimization stage.

**Other PEFT Methods**    LoRA-based methods and variants can be attributed to one type of PEFT [36] method, which also includes (1) Selective training: BitFit [37]; (2) Soft prompt: Prefix Tuning [38] and P-Tuning [39, 40]; (3) Adapter-based method: Serial Adapter [2] [41] and Parallel Adapter [42]. The latter two types also insert extra trainable modules while keeping the pre-trained matrices frozen like LoRA does.

## 3 RidgeLoRA: Matrix Ridge Enhanced LoRA

### 3.1 Architecture of RidgeLoRA

RidgeLoRA switches the computation graph to the following formalization:

$$X_{out} = X_{in}(\lambda\Sigma + \frac{\alpha}{r}AB)W, \tag{1}$$

---

[2]The biggest difference between RidgeLoRA's series connection and serial adapter is that the training parameters of RidgeLoRA can be absorbed into the original weights after the training is completed, whereas the extra modules inserted by serial adapter cannot be absorbed due to the presence of activation values inside of it.

where $\Sigma$ is a diagonal matrix (referred to as the Ridge) that requires gradient descent. It has $d_{in}$ learnable parameters on the matrix diagonal, with its matrix rank $d_{in}$. $\lambda$ denotes the "Ridge Intensity" which can be updated. Specifically, RidgeLoRA inserts additional trainable ridge beside the matrix and novelly converts the extra modules into series connections, as opposed to the parallel connections used in LoRA variants. Moreover, the newly trained part can also be absorbed back to the original weight as described in the following formalization:

$$W' = (\lambda\Sigma + \frac{\alpha}{r}AB)W, \tag{2}$$

where the trained matrix $W'$ is the product of two full-rank matrices. This absorption recovers the original architecture while endowing the model with brand new optimized weights during inference, which provides the same advantage as LoRA and its variants. Moreover, in order to provide a clear comparison to main-stream tuning methods, we further list the properties of them in Table 1.

## 3.2 Theoretical Analysis of RidgeLoRA

According to the setups of RidgeLoRA, $\lambda\Sigma$ is initialized from a part of a diagonal matrix, thus enabling it to dominate the spectrum of $\lambda\Sigma + \frac{\alpha}{r}AB$ and achieve a high rank of $d_{in}$. Here we conduct derivations on how closely the enhanced RidgeLoRA architecture can approximate the full-rank weight update $\Delta W$. By introducing this ridge term, we effectively increase the model's rank expressiveness and improve the overall performance.

**Theorem 3.1.** *Let $K \in \mathbb{R}^{d \times d}$ be a rank-$k$ matrix ($k \leq d$), $D \in \mathbb{R}^{d \times d}$ be a diagonal matrix, $M \in \mathbb{R}^{d \times d}$ be an arbitrary matrix. Given that $M$ has a Singular Value Decomposition (SVD) $M = U\Sigma V^\top$. Let $A \in \mathbb{R}^{n(n-1) \times k}$ be a matrix whose rows are indexed by ordered pairs $(p, q)$ where $p, q \in \{1, 2, ..., n\}$ and $p \neq q$. Each row $a_{(pq)} \in \mathbb{R}^{1 \times k}$ of $A$ has entries given by $U_{pj}V_{qj}$ for $j \in \{1, 2, ..., k\}$. Let $c \in \mathbb{R}^{n(n-1) \times 1}$ be a vector whose entries are $c_{pq} = \sum_{j=k+1}^{n} U_{pj}s_jV_{qj}$, where $s_j$ denotes the corresponding entries in $\Sigma$. We have that*

$$\min_{K,D} \|K + D - M\|_F^2 \leq \|(I - A(A^\top A)^\dagger A^\top)c\|_F^2, \tag{3}$$

*where $(\cdot)^\dagger$ denotes the pseudo inverse of matrix.*

This demonstrates the advantages of adding Ridge. Specifically, the right side of the above expression is less than or equal to the LoRA case. For the detailed proof, please refer to Appendix A.1. Next, we will elaborate on this point.

In the case of LoRA, $D$ is a zero matrix, and the minimization objective becomes

$$\min_K \|K - M\|_F^2,$$

Let $K$ be a rank-$k$ matrix. According to the Eckart–Young–Mirsky theorem [43], the optimal choice $K = U\Sigma_k V^\top$ minimizes the Frobenius norm $\|K - M\|_F^2$. Substituting this optimal $K$, we obtain

$$\|K - M\|_F^2 = \|U(\Sigma - \Sigma_k)V^\top\|_F^2,$$

where $\Sigma_k$ denotes the best rank-$k$ approximation of $\Sigma$, obtained by retaining the top-$k$ largest singular values (ordered in descending magnitude) and zeroing out the rest.

In the proof of Theorem 3.1, after introducing the diagonal matrix $D$, if keeping $K$ fixed as $K = U\Sigma_k V^\top$, the problem reduces to a least squares optimization with respect to the entries of $D$. Clearly, $D = 0$ is not the optimal solution. The bound we established demonstrates that our method is more effective, detailed analysis can be found in Appendix A.1 to illustrate a theoretical measure.

## 3.3 Detailed Designs of RidgeLoRA

On top of the theoretical analysis of how RidgeLoRA facilitates full-rank training, additional mechanisms are introduced to optimize performance. Specifically, it incorporates a novel weight initialization strategy and an auxiliary loss function for the efficient updates of parameters.

**Algorithm 1** Weight Initialization of RidgeLoRA

---

**Input:** Input dimension $d_{in}$, Target low rank $r$, Scaling factor $\alpha$

**Output:** $\lambda, \Sigma, A, B$

1: Initialize $\lambda \leftarrow 1$.  ▷ *Initialize the intensity of ridge term.*

2: Sample noise vector $\mathbf{N} \in \mathbb{R}^r$ using

$$\mathbf{N} = \boldsymbol{\sigma}[\mathcal{N}(\mu_{\mathbf{N}}, \sigma_{\mathbf{N}}^2)],$$

where $\boldsymbol{\sigma}(\cdot)$ is the sigmoid function.  ▷ *Split a small portion to initialize low-rank matrices.*

3: Construct matrix $\Sigma \in \mathbb{R}^{d_{in} \times d_{in}}$ as

$$\Sigma = \begin{bmatrix} \mathrm{diag}(\mathbf{N}) & 0 \\ 0 & I_{d_{in}-r} \end{bmatrix},$$

where $\mathrm{diag}(\mathbf{N})$ is an $r \times r$ diagonal matrix, and $I_{d_{in}-r}$ is an identity matrix.

4: Initialize low-rank matrix $A \in \mathbb{R}^{d_{in} \times r}$ with Gaussian Noise $\mathcal{N}(0, 1/r)$

5: Conduct SVD on matrix $A$: $U_A \Sigma_A V_A^\top = A$.

6: To ensure the condition  ▷ *This ensures the "transform-calibrating restriction".*

$$\lambda\Sigma + \frac{\alpha}{r} AB = I,$$

$B$ is initialized with

$$B = \frac{r}{\alpha} V_A \Sigma_A^{-1} U_A^\top \left( I_{d_{in}} - \lambda\Sigma \right).$$

▷ *Note that $U_A^\top U_A = V_A^\top V_A = I_r$.*

---

**Weight Initialization**  In the field of deep learning, matrices are usually initialized with well-designed methods [44–46] for a better starting point. However, vanilla LoRA is constrained by the *"transform-calibrating restriction"* [12], which allows only the initialization of matrix $A$ with such methods, while $B$ is initialized as an all-zero matrix. This limitation hinders the ability of LoRA to fully explore the parameter space. With the novel series connection, this restriction can be fulfilled with $\lambda\Sigma + \frac{\alpha}{r} AB = I$, eliminating the strong constraint that requires $B = 0$. This ensures good initializations can be adopted on all matrices, endowing them with the possibility to get efficiently updated. In practice, given $AB$'s target low rank $r$, a small portion ($r \times r$) of the identity matrix is initialized with a diagonal matrix $\mathrm{diag}(\mathbf{N})$. This leads to be both part (the Ridge and the low-rank matrices) to be non-zero, allowing for good initialization of $AB$ while ensuring $\lambda\Sigma$ dominates the spectrum. Subsequently, RidgeLoRA initializes the matrix $A$ (with a Gaussian distribution, by default), and then compute matrix $B$ to ensure the sum of these modules equal to the identity matrix. The whole procedure is also demonstrated in Alg. 1 for a clear presentation.

It is noteworthy that this process also differs from SVD-based methods [17, 18] that initialize matrices with the eigenvalues and eigenvectors of the pre-trained matrix. They require massive computation to conduct SVD on the full-rank original matrices and split corresponding portion to initialize the low-rank matrices, while RidgeLoRA only requires SVD on the low-rank matrix $A$, quantitative comparisons can also be found at Table 1.

**Ridge Squashing Loss**  The initialization of the low-rank modules ensures that their sum equals an identity matrix $I$, hereby adhering to the *"transform-calibrating restriction"*. Furthermore, to encourage the matrix to explore a broader space, RidgeLoRA adds a loss term $\mathcal{L}_{\mathbf{RS}}$, pushing the weight update towards higher rank:

$$\mathcal{L}_{\mathbf{RS}} = \frac{1}{LB} \sum_{l=1}^{L} \sum_{b=1}^{B} \left\| \lambda_{l,b}\Sigma_{l,b} + \frac{\alpha}{r} A_{l,b} B_{l,b} - I \right\|_*, \tag{4}$$

where $L$ is the total number of layers in the model, $B$ denotes the number of matrix blocks in each layer, and $\|\cdot\|_*$ is the nuclear norm [47], which measures the rank of the weight updates. Notably, $\beta_{\mathbf{RS}}$ is set to a negative value, which encourages rank updates in the non-diagonal portions, hereby increasing the effectiveness of training the matrices $A$ and $B$. We further incorporate this loss into the original objective:

$$\mathcal{L} = \mathcal{L}_{model} + \beta_{\mathbf{RS}} \mathcal{L}_{\mathbf{RS}}, \tag{5}$$

| Method | ↓ Trainable Parameters (%) | BoolQ (Acc.) | PIQA (Acc.) | SIQA (Acc.) | HellaSwag (Acc.) | WinoGrande (Acc.) | ARC-e (Acc.) | ARC-c (Acc.) | OBQA (Acc.) | Avg. (Acc.) |
|---|---|---|---|---|---|---|---|---|---|---|
| *Llama-2-7B* | | | | | | | | | | |
| FFT | 100% (6.74B) | 71.76 | 83.84 | 80.81 | 92.45 | 83.66 | 82.87 | 72.35 | 84.40 | 81.52 |
| LoRA | 0.30% (20.02M) | 66.92 | 81.66 | 77.43 | 90.68 | 75.53 | 84.01 | 66.55 | 74.40 | 77.15 |
| DoRA | 0.32% (21.37M) | 67.50 | 81.23 | 77.48 | 90.78 | 76.16 | 84.34 | 67.06 | 74.20 | 77.35 |
| PiSSA | 0.30% (20.02M) | 69.81 | **83.68** | 79.94 | **93.60** | 80.51 | 85.94 | 71.42 | 79.20 | 80.51 |
| KaSA | 0.30% (20.05M) | 66.52 | 80.85 | 76.77 | 90.28 | 75.37 | 83.54 | 65.87 | 73.60 | 76.60 |
| RidgeLoRA | 0.29% (19.42M) | **71.43** | 83.08 | **81.27** | 93.24 | **81.53** | **86.66** | **71.76** | **82.00** | **81.37** |
| *Llama-3.1-8B* | | | | | | | | | | |
| FFT | 100% (8.03B) | 70.88 | 83.62 | 79.02 | 90.93 | 83.11 | 85.10 | 75.34 | 83.20 | 81.40 |
| LoRA | 0.26% (21.00M) | 71.67 | 88.08 | 79.94 | 94.87 | 83.82 | 92.89 | 80.20 | 84.80 | 84.53 |
| DoRA | 0.28% (22.37M) | 71.43 | 88.63 | 80.45 | 94.92 | 83.66 | 93.27 | 80.80 | 85.20 | 84.80 |
| PiSSA | 0.26% (21.00M) | 73.56 | **89.39** | 82.65 | 95.16 | **87.61** | **93.73** | 83.36 | **88.00** | 86.68 |
| KaSA | 0.26% (21.03M) | 71.79 | 88.57 | 79.99 | 94.63 | 83.19 | 92.89 | 80.89 | 84.00 | 84.49 |
| RidgeLoRA | 0.26% (21.23M) | **73.90** | 89.34 | **83.67** | **95.79** | 86.98 | 93.52 | **83.87** | 87.00 | **86.76** |
| *Mistral-v0.3-7B* | | | | | | | | | | |
| FFT | 100% (7.25B) | 71.92 | 85.64 | 79.73 | 92.19 | 85.08 | 83.46 | 73.63 | 84.00 | 81.96 |
| LoRA | 0.29% (21.00M) | 74.47 | 90.26 | 82.50 | 96.18 | 87.45 | 92.72 | 82.00 | 88.89 | 86.81 |
| DoRA | 0.31% (22.37M) | 74.96 | 90.32 | 81.93 | 96.43 | 88.56 | 92.72 | 82.25 | 89.60 | 87.10 |
| PiSSA | 0.29% (21.00M) | **75.18** | 90.48 | **82.86** | **96.61** | 87.77 | 93.10 | 82.34 | **91.00** | 87.42 |
| KaSA | 0.29% (21.03M) | 73.74 | 89.61 | 81.53 | 96.21 | 87.61 | 92.47 | 81.66 | 89.20 | 86.50 |
| RidgeLoRA | 0.29% (21.23M) | 74.63 | **91.35** | 81.88 | 96.38 | **88.63** | **93.18** | **82.59** | 90.80 | **87.43** |

Table 2: Accuracy comparison of Llama-2-7B (MHA), Llama-3.1-8B (GQA) and Mistral-v0.3-7B (GQA) with various tuning methods on eight commonsense reasoning datasets. The numbers of trainable parameters of different methods are also included for a clear comparison. The highest accuracy scores achieved by low-rank methods are marked as **Bold**.

where $\mathcal{L}_{model}$ is the task-related loss for the PEFT-enhanced base model, and $\beta_{\mathbf{RS}}$ is a hyper-parameter controlling the influence of the rank-decreasing penalty. Our experimental results and ablation study also indicate that selecting a negative value further improves performance.

## 4 Experiments

### 4.1 Experimental Setup

**Datasets** In order to showcase the validity of RidgeLoRA and demonstrate its good performance. In comparisons with state-of-the-art low rank methods, we conduct comprehensive experiments across different tasks, which are widely utilized for evaluation in previous works: namely, (i) **Commonsense Reasoning**, (ii) **Math & Code Problems** and (iii) **Multi-modal Understanding** tasks. We adopted all of its training split for fine-tuning for fixed number of steps to ensure fair comparisons. Reported metrics are evaluated on the official test splits.

**Baselines** Across all of our experiments, we mainly compare RidgeLoRA with low-rank based tuning methods, namely vanilla LoRA [12], DoRA [16] and SVD-based PiSSA [17] and KaSA [18]. To further demonstrate the comparable performance of RidgeLoRA on par with FFT, we also include FFT in our main results. Details of the experiments can be found at Appendix B.4.

**Detailed Setups** Throughout our experiments, we adopt a cosine learning rate schedule and use AdamW [48] as the optimizer. Unless otherwise specified, all LoRA variants share the same maximum learning rate for a given task. Concretely, we use a learning rate of $2 \times 10^{-5}$ for Math & Code tasks, and $3 \times 10^{-5}$ for Commonsense tasks. The rank of the low-rank matrices is set to $64$ for Math & Code tasks, and $8$ for Commonsense tasks. For multi-modal understanding, we follow the configuration proposed in DoRA [16] for a fair comparison.

### 4.2 Main Experimental Results

**Commonsense Reasoning** As is showcased in Table 2, across eight commonsense reasoning datasets, the average scores achieved by our proposed RidgeLoRA outperforms most of LoRA variants. When choosing Llama-2-7B as the base model, RidgeLoRA outperforms state-of-the-art baseline, i.e., PiSSA by an improvement of 0.86% in the average accuracy. When analyzing each single dataset, we observe that RidgeLoRA surpasses most of the tuning methods including FFT by large margins. For example, RidgeLoRA outperforms LoRA by a 6.0% accuracy improvement on WinoGrande with

Llama-2-7B as the base model, and is generally better on every dataset than KaSA across different base models. From the results we can observe that only in few datasets that RidgeLoRA may not surpass baselines, with the performance drops usually do not exceed 1%.

As for low-rank methods' performance in calibrating FFT, our proposed RidgeLoRA is the closest to the performance of FFT when selecting Llama-2-7B as the base model, which demonstrates the advantages of full-rank training of RidgeLoRA, with a performance drop of only 0.15%. We also observe that when selecting Llama-3.1-8B and Mistral-v0.3-7B as the base model, the performances of FFT are commonly exceeded by low-rank based methods, which are also observed by many previous works [17, 18, 15], that base model may find it harder to converge given limited data in some tasks comparing to low-rank methods.

**Math & Code Problems** We also evaluate RidgeLoRA together with its baselines on math and code problem solving datasets, where models are usually required to generate long-form arithmetic reasoning trace (math) and complete executable program (code), to further test the performance. As is demonstrated in Table 3, RidgeLoRA surpasses almost all of its baselines with different LLMs as base models. Taking the math benchmarks with Llama-2-7B as the base model as an example, except for FFT, RidgeLoRA outperforms all of its low-rank baselines, even outperforms KaSA by 7.26% on GSM8K. Similar results hold for Llama-3.1-8B and Mistral-0.3-7B models, where RidgeLoRA outperforms its baselines by 0.56% in average and even surpasses FFT by an improvement of 2.42% in the result accuracy. In few datasets, RidgeLoRA may get surpassed by SVD-based PiSSA or KaSA, but with an average margin of only 0.30%.

| Method | ↓ Trainable Parameters (%) | GSM8K (Acc.) | MATH (Acc.) | HumanEval (+) (Pass@1) | MBPP (+) (Pass@1) |
|---|---|---|---|---|---|
| *Llama-2-7B* | | | | | |
| FFT | 100% (6.74B) | 66.32 | 17.72 | 37.8 (35.4) | 45.2 (36.8) |
| LoRA | 2.32% (159.9M) | 53.44 | 8.94 | 22.6 (18.3) | 37.0 (**31.0**) |
| DoRA | 2.34% (161.3M) | 52.25 | 8.08 | 24.4 (20.1) | 36.2 (**31.0**) |
| PiSSA | 2.32% (159.9M) | 56.29 | 9.28 | 25.0 (20.1) | 36.8 (29.4) |
| KaSA | 2.33% (160.8M) | 49.18 | 7.20 | 22.0 (19.5) | 35.4 (29.4) |
| RidgeLoRA | 2.13% (147.0M) | **56.44** | 9.76 | **26.2 (23.8)** | **37.3** (29.9) |
| *Llama-3.1-8B* | | | | | |
| FFT | 100% (8.03B) | 77.77 | 28.84 | 58.5 (55.5) | 64.0 (55.6) |
| LoRA | 2.05% (167.8M) | 75.97 | 28.30 | 51.2 (47.0) | 67.5 (56.6) |
| DoRA | 2.06% (169.2M) | 76.05 | 27.82 | 51.2 (48.2) | 67.7 (56.6) |
| PiSSA | 2.05% (167.8M) | 77.92 | **30.26** | 53.7 (50.0) | 65.1 (56.1) |
| KaSA | 2.06% (168.7M) | 75.82 | 27.52 | 53.0 (50.6) | 68.5 (58.2) |
| RidgeLoRA | 1.96% (160.7M) | **78.07** | 30.12 | 53.7 (**51.8**) | **69.8 (59.8)** |
| *Mistral-v0.3-7B* | | | | | |
| FFT | 100% (7.25B) | 68.11 | 21.68 | 49.4 (47.0) | 51.3 (43.1) |
| LoRA | 2.26% (167.8M) | 71.56 | 21.56 | 45.7 (39.0) | 62.2 (51.1) |
| DoRA | 2.28% (169.2M) | 72.83 | 22.08 | 45.1 (39.0) | 60.8 (51.3) |
| PiSSA | 2.26% (167.8M) | 72.53 | 22.96 | 47.6 (40.9) | 62.7 (51.3) |
| KaSA | 2.28% (168.7M) | **74.33** | 23.06 | 47.6 (40.2) | 62.2 (50.5) |
| RidgeLoRA | 2.17% (160.7M) | 73.88 | **24.02** | **48.2 (41.5)** | **63.2 (53.7)** |

Table 3: Performances of different models on math & code problem benchmarks. For the code benchmarks, **(+)** denotes the enhanced datasets with more difficult test cases, whose metrics are in the parentheses, best performances of low-rank methods are marked as **Bold**.

Similar results hold for the code benchmarks. On the datasets with original test cases, RidgeLoRA surpasses every low-rank method and outperforms the best of its baseline performances with a 0.65% improvement. As for datasets with enhanced test cases, i.e., the ones with (+), RidgeLoRA outperforms the low-rank methods by a 1.4% improvement, showcasing its good performance and generalizability on hard cases. Like Commonsense Reasoning, we also observe several low-rank methods even achieve better performance than FFT, the reason is that the base models already possess enough capabilities, thus possible to achieve good results by tuning fewer parameters on the training set with less data.

**Multi-modal Understanding** In order to expand the applicable domain of RidgeLoRA, we further evaluate RidgeLoRA's performance in aligning a pre-trained a language model with a multi-modal projector. The evaluation results can be found at Table 4. As is showcased, it is showcased that language model tuned with RidgeLoRA achieves better performances than DoRA, with an improvement of at most 1.4% comparing to

| Method | ↓ Trainable Parameters (%) | GQA | SQA | VQA$^{\mathrm{T}}$ | POPE | Avg. |
|---|---|---|---|---|---|---|
| FFT | 100% | 61.9 | 66.8 | 58.2 | 85.9 | 68.20 |
| LoRA | 4.61% | **62.9** | 68.4 | 58.2 | 86.4 | 68.98 |
| DoRA | 4.63% | **62.9** | **69.9** | 57.0 | 87.2 | 69.25 |
| RidgeLoRA | **4.26%** | 61.9 | 69.2 | **58.4** | **87.6** | **69.28** |

Table 4: Multi-modal understanding evaluation results of different tuning methods on 4 vision-language tasks following the setups of DoRA [16], best performances are marked **Bold**.

| Method | BoolQ (Acc.) | PIQA (Acc.) | SIQA (Acc.) | HellaSwag (Acc.) | WinoGrande (Acc.) | ARC-e (Acc.) | ARC-c (Acc.) | OBQA (Acc.) | Avg.(C) (Acc.) |
|---|---|---|---|---|---|---|---|---|---|
| **Ridge Enhanced Parallel Connections** | | | | | | | | | |
| *LoRA+Ridge* | $67.53_{(\uparrow 0.61)}$ | $82.37_{(\uparrow 0.71)}$ | $78.25_{(\uparrow 0.82)}$ | $91.77_{(\uparrow 1.09)}$ | $76.40_{(\uparrow 0.87)}$ | $84.97_{(\uparrow 0.96)}$ | $68.00_{(\uparrow 1.45)}$ | $74.40_{(\uparrow 0.00)}$ | $77.96_{(\uparrow 0.81)}$ |
| *DoRA+Ridge* | $67.23_{(\downarrow 0.27)}$ | $82.43_{(\uparrow 1.20)}$ | $78.71_{(\uparrow 1.23)}$ | $92.03_{(\uparrow 0.25)}$ | $76.80_{(\uparrow 0.64)}$ | $85.23_{(\uparrow 0.89)}$ | $68.09_{(\uparrow 1.03)}$ | $76.20_{(\uparrow 2.00)}$ | $78.34_{(\uparrow 0.99)}$ |
| *PiSSA+Ridge* | $70.64_{(\uparrow 0.83)}$ | $83.57_{(\uparrow 1.20)}$ | $80.55_{(\uparrow 0.61)}$ | $93.43_{(\downarrow 0.17)}$ | $80.66_{(\uparrow 0.15)}$ | $86.24_{(\uparrow 0.30)}$ | $71.33_{(\downarrow 0.09)}$ | $80.60_{(\uparrow 1.40)}$ | $80.88_{(\uparrow 0.37)}$ |
| *KaSA+Ridge* | $68.41_{(\uparrow 1.89)}$ | $83.13_{(\uparrow 2.28)}$ | $79.84_{(\uparrow 3.07)}$ | $92.57_{(\uparrow 2.29)}$ | $77.74_{(\uparrow 2.37)}$ | $85.14_{(\uparrow 1.60)}$ | $68.52_{(\uparrow 2.65)}$ | $79.20_{(\uparrow 5.60)}$ | $79.32_{(\uparrow 2.72)}$ |
| **Weight Initialization** | | | | | | | | | |
| *Diagonal* | 66.59 | 80.30 | 76.61 | 91.03 | 75.14 | 83.96 | 64.33 | 73.40 | 76.42 |
| *Gaussian*(default) | **71.43** | **83.08** | **81.27** | 93.24 | 81.53 | **86.66** | 71.76 | 82.00 | 81.37 |
| *Kaiming* ($\mathcal{N}$) | 70.88 | **83.08** | 80.50 | **93.77** | 82.00 | 85.48 | **72.10** | **83.80** | **81.45** |
| *Kaiming* ($\mathcal{U}$) | 70.02 | 82.48 | 79.48 | 93.67 | **82.40** | 85.65 | 71.59 | 81.60 | 80.86 |
| *Xavier* ($\mathcal{N}$) | 68.63 | 82.04 | 78.35 | 91.98 | 78.22 | 85.10 | 68.09 | 76.00 | 78.55 |
| *Xavier* ($\mathcal{U}$) | 67.99 | 82.21 | 78.45 | 91.86 | 77.74 | 85.44 | 68.43 | 75.20 | 78.42 |

Table 5: Experimental results of Commonsense Dataset from two ablation studies, namely: (1) Parallel Connection; (2) Weight Initialization. $\mathcal{N}$ denotes the normal distribution while $\mathcal{U}$ is the uniform distribution of corresponding method. **Avg.(C)** denotes the average accuracy score of Commonsense datasets. The subscripts represent the performance change of adding the Ridge enhancement compared to itself. Best scores of initialization methods are marked **Bold**.

DoRA on **VQA**[T]. Consistent with the training of language-only models, the ratio of trainable parameters remains lower than low-rank baselines, demonstrating the lightweight advantages of RidgeLoRA.

### 4.3 Ablation Study

Apart from comprehensive evaluation with various types of model and different tasks comparing with baselines, which provides solid evidences that the RidgeLoRA surpasses vanilla LoRA together with its state-of-the-art variants, we also conduct ablation study to demonstrate the validity of different parts of RidgeLoRA at a fine-grained level.

**Ridge Enhanced Parallel Connections** Since we modify the connection between the newly trained modules and the original matrix, in order to provide in-depth analysis about the necessity of series connection, we also conduct ablation study with different LoRA variants enhanced with a Ridge alongside. The experimental results can be found in the first group of Table 5 and Table 6, where performance improvements can be observed on each method comparing to itself without the Ridge. On the average score of Commonsense Dataset, KaSA is observed to get enhanced with Ridge by a 2.72% improvement. PiSSA is also enhanced, with an average improvement of 0.37% on Commonsense and a 0.18% improvement on math datasets. These results prove that RidgeLoRA can better approximate a full-rank weight update, thus leading to better performances. It is noteworthy that here we prove that the series connection RidgeLoRA not only surpasses the parallel connection, but also outperforms the other LoRA variants enhanced by Ridge, validating the effectiveness of series connection.

| Method | GSM8K (Acc.) | MATH (Acc.) |
|---|---|---|
| *LoRA+Ridge* | $55.39_{(\uparrow 1.95)}$ | $9.14_{(\uparrow 0.20)}$ |
| *DoRA+Ridge* | $54.12_{(\uparrow 1.87)}$ | $8.72_{(\uparrow 0.64)}$ |
| *PiSSA+Ridge* | $56.36_{(\uparrow 0.07)}$ | $9.56_{(\uparrow 0.28)}$ |
| *KaSA+Ridge* | $51.42_{(\uparrow 2.24)}$ | $7.98_{(\uparrow 0.78)}$ |

Table 6: Performances on math datasets existing LoRA variants enhanced with parallel Ridge.

**Weight Initialization Methods** Here we remain the series connection setup and further compare the performances of different initialization methods of matrix $A$, namely *Gaussian* [44], *Xavier* [45], and *Kaiming* [46] initialization, the results can be found at the second group in Table 5 and Table 7. We also include an initialization where matrix $A$ only has diagonal values before training, termed as *Diagonal*. From the results we can observe that randomized matrix initialization always leads to better performances, which is demonstrated by the fact that *Diagonal* gets surpassed by others by a margin of at least 0.78%. As for the normal distribution initialized methods, we can observe that *Gaussian*, with a higher variance $\frac{1}{r}$ in our default setup, has better performances than *Kaiming* and *Xavier*,

| Method | GSM8K (Acc.) | MATH (Acc.) |
|---|---|---|
| *Diagonal* | 54.04 | 8.98 |
| *Gaussian*(default) | **56.44** | **9.76** |
| *Kaiming* ($\mathcal{N}$) | 56.14 | 9.62 |
| *Kaiming* ($\mathcal{U}$) | 55.84 | 9.46 |
| *Xavier* ($\mathcal{N}$) | 55.46 | 9.08 |
| *Xavier* ($\mathcal{U}$) | 55.09 | 8.92 |

Table 7: Performances on math datasets of RidgeLoRA with different initializations.

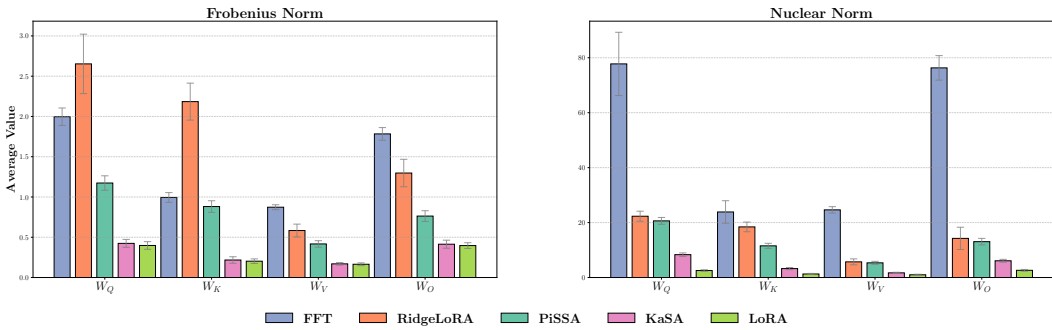

Figure 2: Analysis on the weight update patterns of the Self-Attention module of Llama-3.1-8B (GQA). Norm here denotes the norm of weight updates, error lines come from variances between different layers.

with a 0.64% improvement on GSM8K. Moreover, initializing with uniform distributions won't lead to good performances when compared to normal distributions, their performance drop can achieve up to 2.20% on OBQA if taking *Kaiming (U)* as an example.

**Pattern Analysis of Weight Updates** As an auxiliary analysis, we also performed a visualization of the equivalent parameter updates of different methods. As is illustrated in Figure 2, constrained with the low-rank nature, the update norms of other LoRA variants are always lower than those of FFT, especially the vanilla LoRA. Benefit from the full-rank nature of RidgeLoRA, its updates are comparable or sometimes larger than that of FFT, showcasing its good representation capability.

**Ridge Squashing Loss Fusion** As we fuse Ridge Squashing Loss with the original loss to facilitate the training process, in order to prove its validity, we conduct ablation study with different hyper-parameter setups. From the evaluation results from Table 8, we can observe that on all eight datasets, a negative value of $\beta_{\mathbf{RS}}$ tends to lead to better model performances, with a gap of at most 1.21% in average score across two groups. Comparing to the baseline group (w/o $\mathcal{L}_{\mathbf{RS}}$), selecting a negative values always bring about better performance, with its scores all above baseline accuracy. Furthermore, we conduct Student's t-test [49] to showcase that selecting a negative value for $\beta_{\mathbf{RS}}$ significantly improves the performance, which means increasing the rank of weight updates is favorable for good performances. Details can be found at Appendix B.1.

## 5 Conclusion

In this paper, we propose RidgeLoRA, a novel PEFT approach that enhances the vanilla LoRA architecture through two key innovations: replacing the conventional parallel structure with series connection and incorporating a diagonal ridge term. Combined with the well designed initialization strategy and ridge squashing loss, RidgeLoRA achieves superior capability while maintaining computational efficiency. Through rigorous theoretical analysis and comprehensive experimental evaluations, we demonstrated that RidgeLoRA consistently outperforms existing approaches. Our work opens up new possibilities for PEFT and provides valuable insights into PEFT methods.

## 6 Limitations

While RidgeLoRA is supported by comprehensive experiments and rigorous mathematical derivations. However, several limitations remain: 1) Its potential in scenarios such as continual learning [50, 51] and model editing [52] where vanilla LoRA is commonly applied has not yet been explored. Future work could investigate how RidgeLoRA extends to these settings and whether it offers advantages in such contexts. 2) Due to computational constraints, we do not include experiments on very large models (e.g., those with more than 70B parameters). Nonetheless, the reported results, along with detailed ablations, provide a compelling demonstration of the method's effectiveness on a range of models.

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

# A Theorems and Math Derivations

## A.1 Proof of Theorem 3.1

**Theorem A.1.** *Let $K \in \mathbb{R}^{d \times d}$ be a rank-$k$ matrix ($k \leq d$), $D \in \mathbb{R}^{d \times d}$ be a diagonal matrix, $M \in \mathbb{R}^{d \times d}$ be an arbitrary matrix. Given that $M$ has a Singular Value Decomposition (SVD) $M = U\Sigma V^{\top}$. Let $A \in \mathbb{R}^{n(n-1) \times k}$ be a matrix whose rows are indexed by ordered pairs $(p, q)$ where $p, q \in \{1, 2, ..., n\}$ and $p \neq q$. Each row $a_{(pq)} \in \mathbb{R}^{1 \times k}$ has entries given by $U_{pj} V_{qj}$ for $j \in \{1, 2, ..., k\}$. Let $c \in \mathbb{R}^{n(n-1) \times 1}$ be a vector whose entries are $c_{pq} = \sum_{j=k+1}^{n} U_{pj} s_j V_{qj}$, where $s_j$ denotes the corresponding entries in $\Sigma$. We have that*

$$\min_{K,D} \|K + D - M\|_F^2 \leq \|(I - A(A^{\top}A)^{\dagger}A^{\top})c\|_F^2 \tag{6}$$

*Proof.* We construct an upper-bound objective function via a relaxation of the original problem. Perform the singular value decomposition (SVD) of $M$ as $M = U\Sigma V^{\top}$. Define $K = U\widetilde{\Sigma}_k V^{\top}$, where $\widetilde{\Sigma}_k$ is a rank-$k$ diagonal matrix which is left for optimization. Then we have

$$\min_{K,D} \|K + D - M\|_F^2 \leq \min_{\widetilde{\Sigma}_k, D} \|U\widetilde{\Sigma}_k V^{\top} + D - U\Sigma V^{\top}\|_F^2$$

$$= \min_{\widetilde{\Sigma}_k, D} \|D - U(\Sigma - \widetilde{\Sigma}_k)V^{\top}\|_F^2, \tag{7}$$

which serves as a tractable upper bound of the original objective.

Subsequently, we conduct two steps to find the upper bound of (7). The first step is to get the analystic value of $D$ to minimize (7) while fixing $\widetilde{\Sigma}_k$. Then $D$ can be represented in terms of $\widetilde{\Sigma}_k$. In the second step, we substitute $D$ with its expression in terms of $\widetilde{\Sigma}_k$, and then minimize (7) with respect to $\widetilde{\Sigma}_k$.

***Derivation of the optimal $D$.*** Fix $\widetilde{\Sigma}_k$, here we denote $A = U(\Sigma - \widetilde{\Sigma}_k)V^{\top}$. We need to minimize $\|D - A\|_F^2$ with respect to $D$. Since $D$ is diagonal, i.e., $D_{ij} = 0$ where $i \neq j$. We have that

$$\|D - A\|_F^2 = \sum_{i,j}(D_{i,j} - A_{i,j})^2$$

$$= \sum_i (D_{i,i} - A_{i,i})^2 + \sum_{i \neq j} A_{i,j}^2.$$

The objective above is minimized when $D_{i,i} = A_{i,i}$. Therefore, problem (7) reduces to minimizing

$$\sum_{i \neq j} \left(U(\Sigma - \widetilde{\Sigma}_k)V^{\top}\right)_{i,j}^2,$$

with respect to the rank-$k$ diagonal matrix $\widetilde{\Sigma}_k$.

***Derivation of the optimal $\widetilde{\Sigma}_k$.*** Let $Y = \Sigma - \widetilde{\Sigma}_k$. Since $\Sigma$ and $\widetilde{\Sigma}_k$ are diagonal matrices, $Y$ is also a diagonal matrix. Let $\Sigma = \text{diag}(s_1, s_2, ..., s_n)$ and $\widetilde{\Sigma}_k = \text{diag}(t_1, t_2, ..., t_n)$. Note that there are exactly $k$ non-zero diagonal entries in $\widetilde{\Sigma}_k$. Let $J_0 \subset \{1, ..., n\}$ be the set of $n - k$ indices where $t_j = 0$. Let $J_k \subset \{1, 2, ..., n\}$ be the set of $k$ indices where $t_j \neq 0$. The target objective can be rewritten as follows

$$\min_{J_K, \{t_j\}_{j \in J_k}} \sum_{p \neq q} \left([UYV^{\top}]_{pq}\right)^2 = \min_{J_K, \{t_j\}_{j \in J_k}} \sum_{p \neq q} \left(\sum_{j=1}^n U_{pj} y_j V_{qj}\right)^2. \tag{8}$$

For a fixed choice of the index set $J_k$, this is a linear least square problems with respect to $k$ variables $\{t_j\}_{j \in J_k}$.

Let $x$ denote the $k \times 1$ vector whose elements are $s_j - t_j$ for $j \in J_k$. There are $n(n - 1)$ pairs of $(p, q)$ with $p \neq q$. For each pair $(p, q)$, let $a_{(pq), J_k}$ be a $1 \times k$ row vector with entries $U_{pj} V_{qj}$ for

$j \in J_k$, $c_{pq,J_0} = \sum_{j \in J_0} U_{pj} s_j V_{qj}$. Let $A_{J_k}$ be an $n(n-1) \times k$ matrix whose rows are $a_{(pq),J_k}$, and let $c_{J_k}$ be an $n(n-1) \times 1$ vector with entries $c_{pq,J_0}$.

For a fixed index set $J_k$, the minimization objective (8) can be rewritten as

$$\min_{J_k,x} \|A_{J_k}x + c_{J_k}\|_F^2.$$

The solution to this least squares problem is given by $x^* = -(A_{J_k}^\top A_{J_k})^\dagger A_{J_k}^\top c_{J_k}$. The minimum value for a fixed $J_k$ is

$$G(J_k) = \|(I - A_{J_k}(A_{J_k}^\top A_{J_k})^\dagger A_{J_k}^\top)c_{J_k}\|_F^2.$$

Each choice of $J_k$ is an upper bound of the target objective. The final step is to find the optimal set of $k$ indices $J_k$ that minimize $G(J_k)$. There are $\binom{n}{k}$ possible choices for the set $J_k$. The overall minimum value of the above objective function is

$$\min_{J_k \subset \{1,2...,n\}, |J_k|=k} \|(I - A_{J_k}(A_{J_k}^\top A_{J_k})^\dagger A_{J_k}^\top)c_{J_k}\|_F^2. \tag{9}$$

The upper bound in the theorem can be obtained by setting $J_k = \{1, 2, ..., k\}$.

$\square$

## A.2 Weight Updates of FFT

Here we present two perspectives on the weight updates during FFT. While low-rank matrices are usually used to approximate full-rank matrices, we propose enhancing this approximation with a ridge term, which provides better calibration to the original full-rank update scheme. There are two equivalent ways to view the weight updates of the original matrix $W$:

- Additive update (Parallel connection): $W + \Delta W$;

- Multiplicative update (series connection): $W \cdot \Delta W'$.

The full equivalence of additive update and full-rank training is derived further at Appendix A.5. If freezing the original matrix $W$, these two representations are equivalent, as we can establish a bi-directional mapping between them:

- Given any additive update $\Delta W$, we can find its multiplicative counterpart as $\Delta W' = W^{-1}(W + \Delta W)$;

- Given any multiplicative update $\Delta W'$, we can find its additive counterpart as $\Delta W = W \cdot \Delta W' - W$.

In both cases, theoretically $W + \Delta W = W \cdot \Delta W'$. It is noteworthy that vanilla LoRA can be viewed as a low-rank approximation to the parallel connection, where $\Delta W$ is constrained to be the product of two low-rank matrices. We will also demonstrate the experimental results of series and parallel connections with the setup of RidgeLoRA.

## A.3 Singular Value Decomposition with Math Bounded Error

According to Eckart-Young-Mirsky Theorem [43], when conducting low-rank decomposition with SVD on a matrix $W$ to obtain a low-rank matrix $W_k$ with rank $k$. The decomposition error, calculated with Frobenius Norm [53], is equal to the sum of the squares of the singular values of the compressed part of the matrix, which can be formalized as

$$\|W - W_k\|_F^2 = \sum_{i=k+1}^{\min(d_{in},d_{out})} \sigma_i^2,$$

where $W = U\Sigma V^T$ and $W_k = U\Sigma_k V^\top$, $\sigma_i$ is the singular value calculated with SVD, rearranged by descending order. This theorem also gives a math bound to the error of a low-rank decomposition approximating the original full-rank matrix, any matrix $W_k$ with rank $k$ won't better approximate the matrix $W$. However, if breaking the low-rank restrictions with a full-ranked ridge, though with limited parameters, the approximation error gets way lower compared to SVD-based methods. This justifies the necessity of restoring the rank of low-parameter modules to gain better performances.

## A.4 Weight Updates of Vanilla LoRA

LoRA achieves efficient fine-tuning with the low-rank matrices, which is particularly effective when the target task requires capabilities that are closely related to those acquired during pre-training, necessitating only minimal weight adjustments. In the implementation of vanilla LoRA [12], an input signal $X_{in}$ is computed in the following way:

$$X_{out} = X_{in}(W + \frac{\alpha}{r}AB), \tag{10}$$

where $A \in \mathbb{R}^{d_{in} \times r}, B \in \mathbb{R}^{r \times d_{out}}$, $r$ denotes the preset low rank of the two matrices and $\alpha$ acts as the scaling hyper-parameter. The low-rank setup freezes the original matrix $W$, making the weights to be updated lightweight. The architecture enables the extra matrices: $A$ and $B$ to be absorbed by the pre-trained weights, thus feasible to reorganized into a model with exactly the same structure as the original one. The weights can be reorganized as follows:

$$W' = W + \frac{\alpha}{r}AB. \tag{11}$$

Here we take a look at the weight update $\Delta W$, it is an item-wise weight difference between fine-tuned weight $W'$ and the original weight $W$:

$$\Delta W = W' - W = \frac{\alpha}{r}AB, \tag{12}$$

where $A$ and $B$ are low-rank matrices, making this module short of representation capabilities. According to Biderman et al. [14], Shuttleworth et al. [15], LoRA can not fully calibrate full-rank training and may entail *intruder dimensions* for the existence of low-rank matrices.

## A.5 Gradient Analysis of Full-Rank Training

Figure 3: Schematic of matrix computation with input matrix $X_{in}$ and output $X_{out}$.

Here we analyze the matrix computation in detail and the difference between the weight updates of FFT and LoRA training. Specifically, as is depicted in Figure 3, $X_{out}$ is obtained by matrix computation from $X_{in}$, which results in the following computation path:

$$\begin{cases} X_{out} = X_{in}W \\ \dfrac{\partial X_{out}}{\partial W} = X_{in}^{\top} \end{cases} \tag{13}$$

**Theorem A.2.** *FFT is equivalent to adding a full-rank matrix through parallel connection while freezing the original matrix.*

▷ *Gradient of FFT: Suppose that the gradient $\frac{\partial \mathcal{L}}{\partial X_{out}}$ has been backpropagated [54] from subsequent modules. Our goal is to compute the corresponding gradient w.r.t. the input, $\frac{\partial \mathcal{L}}{\partial X_{in}}$, through applying the chain rule. Specifically, given the transformation $X_{out} = X_{in}W$, the gradient can be calculated as:*

$$g_W = \frac{\partial \mathcal{L}}{\partial W} = \frac{\partial \mathcal{L}}{\partial X_{out}} \cdot \frac{\partial X_{out}}{\partial W} = \frac{\partial \mathcal{L}}{\partial X_{out}}X_{in}^{\top}, \tag{14}$$

*from which we can observe that the gradient is calculated by the product of input activation and gradients from subsequent modules, which can be further utilized by optimizers like AdamW [48] to compute the weight updates.*

▷ *Gradient of an Additive (Parallel) Matrix: Here we derive how parameters of an extra inserted parallel matrix $W'$ update during training, where the computation is transformed to the formalization below:*

$$X_{out} = X_{in}(W + \Delta W), \tag{15}$$

*where $\Delta W$ is full-rank while $W$ is kept frozen during training. The gradient of the inserted matrix can be calculated from:*

$$g_{\Delta W} = \frac{\partial \mathcal{L}}{\partial \Delta W} = \frac{\partial \mathcal{L}}{\partial X_{out}} \cdot \frac{\partial X_{out}}{\partial (W + \Delta W)} \cdot \frac{\partial (W + \Delta W)}{\partial \Delta W}$$
$$= \frac{\partial \mathcal{L}}{\partial X_{out}} \cdot \frac{\partial X_{out}}{\partial (W + \Delta W)} \cdot I = \frac{\partial \mathcal{L}}{\partial X_{out}} X_{in}^{\top},$$

(16)

*where the gradient is exactly the same as conducting full-rank training. If the original matrix $W$ is frozen during training while the inserted matrix is initialized as $(\mathbf{0})^{d_{in} \times d_{out}}$ to ensure the output at starting point calibrate the original outputs, every optimization step of parallel added matrix is exactly the same as FFT, hereby leading to the same results.*

## B Details and Analysis of Experiments

### B.1 Hypothesis Test of $\beta_{\mathrm{RS}}$ Selection

In order to complement the analysis of our ablation study in Section 4.3, we conduct t-test [49] on the accuracy scores to analyze the impact of $\beta_{\mathrm{RS}}$ and compare the statistical significance to prove the validity of the proposed ridge squashing loss. Data points across different datasets and $\beta_{\mathrm{RS}}$ values are from Table 8. In the hypothesis test, we refer to $\mu_1$ and $\mu_2$ as the means of the two experimental groups, negative value group and positive value group, respectively, while $\mu_0$ represents a reference data point in which ridge squashing loss is not in effect for comparison in our t-test. The results of the hypothesis tests, including the t-statistics and p-values for each dataset, are summarized in Table 9.

| $\beta_{\mathrm{RS}}$ | BoolQ (Acc.) | PIQA (Acc.) | SIQA (Acc.) | HellaSwag (Acc.) | WinoGrande (Acc.) | ARC-e (Acc.) | ARC-c (Acc.) | OBQA (Acc.) | Avg. (Acc.) |
|---|---|---|---|---|---|---|---|---|---|
| | | | | **Negative Values** | | | | | |
| $-1.0$ | 71.12 | 83.68 | 80.81 | 93.22 | 79.87 | 85.82 | 73.21 | 81.40 | 81.14 |
| $-5 \times 10^{-1}$ | 71.43 | 83.08 | 81.27 | 93.24 | 81.53 | 86.66 | 71.76 | 82.00 | 81.37 |
| $-2 \times 10^{-1}$ | 70.55 | 82.70 | 80.50 | 93.59 | 81.22 | 86.28 | 70.90 | 81.80 | 80.94 |
| $-1 \times 10^{-1}$ | 71.40 | 83.41 | 80.81 | 93.54 | 81.45 | 86.28 | 72.01 | 81.60 | 81.31 |
| $-5 \times 10^{-2}$ | 71.00 | 83.62 | 80.66 | 93.59 | 82.16 | 85.98 | 70.82 | 80.80 | 81.08 |
| $-2 \times 10^{-2}$ | 70.94 | 83.68 | 80.30 | 93.43 | 80.51 | 87.04 | 72.44 | 81.60 | 81.24 |
| $-1 \times 10^{-2}$ | 71.73 | 83.03 | 82.29 | 93.37 | 80.03 | 86.36 | 72.18 | 81.40 | 81.30 |
| $-5 \times 10^{-3}$ | 71.43 | 82.81 | 80.55 | 93.52 | 80.19 | 86.20 | 72.35 | 81.80 | 81.11 |
| $-1 \times 10^{-3}$ | 70.82 | 83.19 | 80.55 | 93.26 | 79.87 | 85.82 | 71.08 | 79.20 | 80.47 |
| $0$ (w/o $\mathcal{L}_{\mathrm{RS}}$) | 70.61 | 82.21 | 80.25 | 93.49 | 80.35 | 85.77 | 71.33 | 82.20 | 80.78 |
| | | | | **Positive Values** | | | | | |
| $1 \times 10^{-3}$ | 70.61 | 82.21 | 80.04 | 92.71 | 80.03 | 85.48 | 72.01 | 79.60 | 80.34 |
| $5 \times 10^{-3}$ | 70.58 | 82.64 | 80.25 | 92.60 | 80.82 | 86.28 | 71.84 | 79.80 | 80.60 |
| $1 \times 10^{-2}$ | 71.12 | 82.54 | 80.04 | 92.83 | 80.74 | 85.56 | 71.33 | 80.8 | 80.62 |
| $2 \times 10^{-2}$ | 70.64 | 82.75 | 80.19 | 92.99 | 80.82 | 85.35 | 72.53 | 80.40 | 80.71 |
| $5 \times 10^{-2}$ | 70.42 | 81.66 | 80.45 | 92.73 | 80.82 | 85.56 | 71.84 | 80.80 | 80.54 |
| $1 \times 10^{-1}$ | 70.51 | 82.10 | 79.94 | 92.84 | 80.03 | 85.77 | 71.59 | 81.00 | 80.47 |
| $2 \times 10^{-1}$ | 71.15 | 81.77 | 80.14 | 92.91 | 79.87 | 85.56 | 71.25 | 80.20 | 80.36 |
| $5 \times 10^{-1}$ | 70.30 | 81.72 | 79.68 | 92.73 | 80.19 | 85.19 | 70.82 | 80.60 | 80.15 |
| $1.0$ | 69.81 | 82.21 | 80.40 | 92.79 | 80.27 | 84.93 | 71.67 | 79.80 | 80.24 |

Table 8: Ablation study with different $\beta_{\mathrm{RS}}$ values with Llama-2-7B as base model on Commonsense Datasets. Hyper-parameters, except for $\beta_{\mathrm{RS}}$, are set to align with the main experiment. We evaluate across positive and negative in comparison with the $\beta_{\mathrm{RS}} = 0$ test, i.e., the one without $\beta_{\mathrm{RS}}$, to obtain conclusions about how the loss takes effect. Observations from metrics above can be used for hypothesis test.

**Test 1: Will fusing losses by a negative $\beta_{\mathrm{RS}}$ lead to better performance?**

Here we conduct t-test with accuracy scores from negative value group and $\mu_0$ to prove the validity of fusing losses by a negative $\beta_{\mathrm{RS}}$. This is a right-tailed mean test, the null hypothesis and the alternative hypothesis can be expressed as

$$H_0 : \mu_1 \leq \mu_0 \quad \text{(Null Hypothesis)}$$
$$H_a : \mu_1 > \mu_0 \quad \text{(Alternative Hypothesis)}$$

where the test statistic is given by $t_1 = \frac{\bar{x}_1 - \mu_0}{S_1/\sqrt{n_1}}$ where $n$ denotes the number of data points and $S^2$ is the variance, $\bar{x}$ is the mean of all data points from the same group. The null hypothesis is rejected if $t_1 > t_{\alpha, n_1 - 1}$, i.e., $p < \alpha$, where the degree of freedom $df_1$ is $n_1 - 1$. We set the significance level at $p < 0.10$. From the left of Table 9, the differences are significant across five of eight datasets, which proves the validity of $\mathcal{L}_{\mathbf{RS}}$ and indicates that setting $\beta_{\mathbf{RS}}$ a negative value will improve the performance.

| Dataset | $t_1$ $(\mu_1 > \mu_0)$ | $p_1$-value | Significant | Dataset | $t_2$ $(\mu_2 < \mu_1)$ | $p_2$-value | Significant |
|---|---|---|---|---|---|---|---|
| BoolQ | 4.4488 | 0.0021 | ✓ | BoolQ | -3.2042 | 0.0056 | ✓ |
| PIQA | 8.3350 | 0.0000 | ✓ | PIQA | -5.8022 | 0.0000 | ✓ |
| SIQA | 3.0429 | 0.0160 | ✓ | SIQA | -3.4093 | 0.0063 | ✓ |
| HellaSwag | -1.4335 | 0.8104 | ✗ | HellaSwag | -9.8413 | 0.0000 | ✓ |
| Winogrande | 1.4475 | 0.0912 | ✓ | Winogrande | -1.1541 | 0.2722 | ✗ |
| ARC-Easy | 3.8073 | 0.0052 | ✓ | ARC-Easy | -4.1315 | 0.0008 | ✓ |
| ARC-Challenge | 1.9867 | 0.0822 | ✓ | ARC-Challenge | -0.6627 | 0.5188 | ✗ |
| OpenBookQA | -3.1967 | 0.9937 | ✗ | OpenBookQA | -2.8795 | 0.0129 | ✓ |

Table 9: Hypothesis test results for $\mu_1 > \mu_0$ (left) and $\mu_2 < \mu_1$ (right). Across six out of eight datasets, the average accuracy scores from the negative group $\mu_1$ is significantly greater than $\mu_0$. Meanwhile, $\mu_1$ is significantly higher than $\mu_2$ in six out of eight datasets, showcasing the necessity of selecting a negative $\beta_{\mathbf{RS}}$.

**Test 2: Will fusing losses by a positive $\beta_{\mathbf{RS}}$ worsen the performance compared to the negative group?**

In order to fully perform ablation study to facilitate hyper-parameter selection, we also conduct Welch's t-test [55] to make a comparison between the positive and negative group for the uncertainty in the variance of the two groups. This is a left-tailed mean test, where the hypotheses can be expressed as

$$H_0 : \mu_2 \geq \mu_1 \quad \text{(Null Hypothesis)}$$
$$H_a : \mu_2 < \mu_1 \quad \text{(Alternative Hypothesis)}$$

In this case, using the Welch-Satterthwaite formula [56, 55], the $t$ value and the degree of freedom $df$ are calculated as follows:

$$
\begin{aligned}
t_2 &= \frac{\bar{x}_2 - \bar{x}_1}{\sqrt{\frac{S_2^2}{n_2} + \frac{S_1^2}{n_1}}}, \\
df_2 &= \frac{\left(\frac{S_1^2}{n_1} + \frac{S_2^2}{n_2}\right)^2}{\frac{(S_1^2/n_1)^2}{n_1 - 1} + \frac{(S_2^2/n_2)^2}{n_2 - 1}}.
\end{aligned}
\tag{17}
$$

The null hypothesis is rejected if $t_2 < -t_{\alpha, df}$, i.e., $p < \alpha$. From the right table of Table 9, six out of eight datasets are witnessed with a high significance, which illustrates selecting a negative $\beta_{\mathbf{RS}}$ can be much better than picking a positive value for it. Furthermore, this proves that increasing the rank of the non-diagonal part of the weight update is more beneficial to obtain better performances.

### B.2 Results of Natural Language Understanding Benchmark

We also evaluate RidgeLoRA with BERT [57]-like encoder models on **Natural Language Understanding** datasets from the GLUE [58] benchmark. We selected encoder-based discriminative models, RoBERTa [59] and DeBERTaV3 [60], to evaluate RidgeLoRA on NLU tasks on the GLUE benchmarks. Here we present the results from the NLU datasets in Table 10.

From the results in the table, we can observe that the performance improvements hold for small encoder models whose model sizes are usually less than 1B. The Ridge-enhanced model achieved the best results in almost all of the datasets, with its average scores also the highest amongst its baselines. On datasets where Ridge doesn't perform that well, the accuracy drop compared to other well-performed LoRA variants is usually less than 1%, showcasing the scalability and good performance of RidgeLoRA.

| Method | MNLI (Acc.) | SST-2 (Acc.) | MRPC (Acc.) | CoLA (Mcc.) | QNLI (Acc.) | RTE (Acc.) | STS-B (Pcc.) | QQP (Acc.) | Avg. |
|---|---|---|---|---|---|---|---|---|---|
| *RoBERTa-base* | | | | | | | | | |
| *FFT* | 87.6 | 94.8 | 90.2 | 63.6 | 92.8 | 78.7 | 91.2 | 91.5 | 86.3 |
| *LoRA* | 87.2 | 94.4 | 87.3 | 61.8 | 92.9 | 78.0 | 90.8 | **90.7** | 85.4 |
| *DoRA* | **87.4** | 94.4 | 87.5 | 61.6 | **93.1** | **78.3** | 90.6 | 87.9 | 85.1 |
| *PiSSA* | 87.1 | 94.0 | 88.0 | **62.8** | 92.2 | 74.4 | 90.3 | 89.3 | 84.8 |
| *KaSA* | 87.0 | 94.4 | 88.7 | 61.9 | 92.4 | 77.3 | **90.9** | 90.6 | 85.4 |
| *Ridge*[†] | 87.3 | **94.4** | 88.1 | 62.2 | 92.8 | 77.9 | 90.8 | **90.7** | **85.5** |
| *RoBERTa-large* | | | | | | | | | |
| *FFT* | 90.2 | 96.4 | 90.9 | 68.0 | 94.7 | 86.6 | 92.4 | 92.2 | 88.9 |
| *LoRA* | 90.8 | 96.2 | 89.8 | 65.1 | 94.7 | 85.2 | 91.6 | 91.8 | 88.2 |
| *DoRA* | 90.8 | 96.2 | 90.0 | 64.6 | 94.7 | 83.8 | **91.7** | 91.7 | 87.9 |
| *PiSSA* | 90.8 | 96.0 | 90.1 | 66.3 | 94.7 | 85.9 | 91.4 | 91.5 | 88.3 |
| *KaSA* | 89.7 | 96.1 | 90.7 | 67.3 | **94.9** | 84.8 | 91.2 | 91.8 | 88.3 |
| *Ridge*[†] | **90.9** | 96.3 | **91.1** | **68.3** | 94.8 | **86.2** | 91.4 | **92.0** | **88.9** |
| *DeBERTa-v3-base* | | | | | | | | | |
| *FFT* | 90.7 | 96.0 | 88.7 | 69.0 | 93.7 | 86.2 | 90.8 | 92.5 | 88.5 |
| *LoRA* | 89.5 | 95.5 | 89.0 | 68.2 | 94.0 | 84.5 | 90.1 | 91.2 | 87.8 |
| *DoRA* | 89.6 | 95.4 | 89.0 | 68.2 | 94.0 | 84.5 | 90.3 | 87.2 | 87.3 |
| *PiSSA* | **89.7** | 95.5 | 88.5 | **71.6** | 93.8 | 83.4 | 90.6 | **91.5** | 88.1 |
| *KaSA* | 89.6 | 95.3 | **89.5** | 68.7 | 94.1 | 84.8 | 90.5 | 91.0 | 87.9 |
| *Ridge*[†] | **89.7** | **96.1** | 88.5 | 70.2 | **94.3** | **85.0** | 90.7 | 91.4 | **88.2** |
| *DeBERTa-v3-large* | | | | | | | | | |
| *FFT* | **91.8** | 96.9 | 92.2 | 75.3 | **96.0** | 92.7 | 93.0 | 93.0 | 91.4 |
| *LoRA* | **91.8** | 95.8 | 89.7 | 71.9 | 95.7 | 90.5 | 92.1 | 91.6 | 89.9 |
| *DoRA* | 91.7 | 95.8 | 89.5 | 71.7 | 95.7 | 91.2 | 92.0 | 91.8 | 89.9 |
| *PiSSA* | 91.6 | 95.9 | 88.5 | 71.3 | 95.8 | 91.5 | 92.2 | 91.9 | 89.8 |
| *KaSA* | 91.7 | 95.7 | 88.6 | 71.7 | 95.7 | 92.1 | 91.8 | **92.4** | 90.0 |
| *Ridge*[†] | 91.7 | **96.0** | 90.2 | 72.2 | **96.0** | 92.5 | 92.3 | 92.0 | **90.4** |

Table 10: Performance of NLU tasks with BERT-like models. The best results for each model on each dataset are highlighted with a **Gray** background, and the highest scores achieved by low-rank methods are marked as **Bold**. [†] denotes a parallel connected module with Ridge.

## B.3 Resource Consumption of LoRA Variants

In addition to the theoretical analysis as presented in Table 1, we also conduct real-world test of different LoRA variants on its memory cost and time consumed for initialization. The results are as reported in Table 11. As is depicted, the proposed RidgeLoRA hardly increases the computation and memory requirements during training. Consistent with the theoretical results, conducting SVD on original weights in order to initialize LoRA weights (like PiSSA or KaSA) takes way more time than LoRA, DoRA and RidgeLoRA. DoRA always consumes more FLOPs and memory for its decoupling the updates of magnitude and direction.

## B.4 Details Descriptions of Experiments

**Base Models** As we select language generation tasks for evaluation, we choose our base models from state-of-the-art LLMs and select Llama-2-7B [4], Llama-3.1-8B [6] and Mistral-v0.3-7B [5] models. The reason for this selection is that we want to confirm the validity of RidgeLoRA on more models. Also, since the Llama-2 model follows the Multi-Head Attention (61, MHA) architecture, unlike the Grouped-Query Attention (62, GQA) that is adopted by the latter two models, it is beneficial to further prove the generalizability of our effects considering this variation in the architecture. [3] For multi-modal understanding, following DoRA's [16] setup, we adopt vicuna-7b-1.5 [63] as

---

[3]It is noteworthy that following the experimental setups of previous LoRA variants, we have chosen the base versions of the models (not the instruction or the chat version) in all of our experiments, from which we assessed the capability of different tuning methods.

| Method | Rank | Total Parameters | Trainable Parameters | Weight Memory/MB | Computation/FLOPs | Initialization Time/s |
|---|---|---|---|---|---|---|
| *Llama-2-7B* | | | | | | |
| *LoRA* | 8 | 6,758,404,096 | 19,988,480 | 13556.79 | 14,121,986,818,048 | 0.7296 |
| *DoRA* | 8 | 6,759,763,968 | 21,348,352 | 13562.22 | 14,339,184,656,384 | 0.9264 |
| *PiSSA* | 8 | 6,758,404,096 | 19,988,480 | 13556.79 | 14,121,986,818,048 | 269.7828 |
| *KaSA* | 8 | 6,758,418,432 | 20,002,816 | 13556.79 | 14,122,016,178,176 | 269.4534 |
| *RidgeLoRA* | 8 | 6,757,773,536 | 19,357,920 | 13554.26 | 14,118,362,939,392 | 1.4136 |
| *LoRA* | 64 | 6,898,323,456 | 159,907,840 | 14116.46 | 14,408,541,667,328 | 3.0530 |
| *DoRA* | 64 | 6,899,683,328 | 161,267,712 | 14121.90 | 16,146,124,374,016 | 3.2561 |
| *PiSSA* | 64 | 6,898,323,456 | 159,907,840 | 14116.46 | 14,408,541,667,328 | 270.2638 |
| *KaSA* | 64 | 6,899,240,960 | 160,825,344 | 14116.46 | 14,410,420,715,520 | 269.9754 |
| *RidgeLoRA* | 64 | 6,885,306,592 | 146,890,976 | 14064.40 | 14,379,550,638,080 | 7.5198 |
| *Llama-3.1-8B* | | | | | | |
| *LoRA* | 8 | 8,051,232,768 | 20,971,520 | 16144.41 | 15,962,246,086,656 | 1.0216 |
| *DoRA* | 8 | 8,052,609,024 | 22,347,776 | 16149.91 | 16,230,681,542,656 | 0.9259 |
| *PiSSA* | 8 | 8,051,232,768 | 20,971,520 | 16144.41 | 15,962,246,086,656 | 200.8085 |
| *KaSA* | 8 | 8,051,247,104 | 20,985,856 | 16144.41 | 15,962,275,446,784 | 200.6371 |
| *RidgeLoRA* | 8 | 8,051,429,600 | 21,168,352 | 16145.20 | 15,960,098,603,008 | 1.6557 |
| *LoRA* | 64 | 8,198,033,408 | 167,772,160 | 16731.61 | 16,262,893,797,376 | 3.0531 |
| *DoRA* | 64 | 8,199,409,664 | 169,148,416 | 16737.12 | 18,410,377,445,376 | 3.3537 |
| *PiSSA* | 64 | 8,198,033,408 | 167,772,160 | 16731.61 | 16,262,893,797,376 | 202.0986 |
| *KaSA* | 64 | 8,198,950,912 | 168,689,664 | 16731.61 | 16,264,772,845,568 | 202.9946 |
| *RidgeLoRA* | 64 | 8,190,890,208 | 160,628,960 | 16703.04 | 16,245,713,928,192 | 7.5247 |
| *Mistral-v0.3-7B* | | | | | | |
| *LoRA* | 8 | 7,268,995,072 | 20,971,520 | 14579.93 | 15,161,234,685,952 | 0.8190 |
| *DoRA* | 8 | 7,270,371,328 | 22,347,776 | 14585.44 | 15,429,670,141,952 | 1.0500 |
| *PiSSA* | 8 | 7,268,995,072 | 20,971,520 | 14579.93 | 15,161,234,685,952 | 201.1598 |
| *KaSA* | 8 | 7,269,009,408 | 20,985,856 | 14579.93 | 15,161,264,046,080 | 202.1278 |
| *RidgeLoRA* | 8 | 7,269,191,904 | 21,168,352 | 14580.72 | 15,159,087,202,304 | 1.6571 |
| *LoRA* | 64 | 7,415,795,712 | 167,772,160 | 15167.14 | 15,461,882,396,672 | 3.0997 |
| *DoRA* | 64 | 7,417,171,968 | 169,148,416 | 15172.64 | 17,609,366,044,672 | 3.2655 |
| *PiSSA* | 64 | 7,415,795,712 | 167,772,160 | 15167.14 | 15,461,882,396,672 | 202.0486 |
| *KaSA* | 64 | 7,416,713,216 | 168,689,664 | 15167.14 | 15,463,761,444,864 | 204.0085 |
| *RidgeLoRA* | 64 | 7,408,652,512 | 160,628,960 | 15138.56 | 15,444,702,527,488 | 8.2065 |

Table 11: Test results of the resources (time, computation and memory) consumed of different LoRA variants. This corroborates the theoretical results from Table 1.

the language model and adopt the CLIP [64]-based vision projector in LLaVA [65] to obtain the representations of images.

**Datasets** We used comprehensive datasets to evaluate the performance of RidgeLoRA in adapting the language models to different down-stream tasks. Starting from where LoRA is most widely used, we perform supervised fine-tuning (SFT, 9) on the **Commonsense Reasoning** datasets and evaluate it on eight corresponding datasets. In order to further evaluate the performances of RidgeLoRA on long-form generation tasks, we also include **Math & Code Problems**, which require the model to have a strong reasoning and generation ability; Furthermore, besides language-only evaluation, we extend our experiments to **Natural Language Understanding** datasets, where language models are aligned to visual projectors to understand images.

**i.** For the Commonsense Reasoning datasets, following previous works, we conduct multi-task training with the training split of eight related datasets, namely **BoolQ** [66], **PIQA** [67], **SocialIQA** [68], **HellaSwag** [69], **WinoGrande** [70], **ARC-Easy**, **ARC-Challenge** [71] and **OpenbookQA** [72]. These datasets require the large models to fully utilize the real-world commonsense knowledge to answer a question.

**ii.** As for Math&Code problems, we evaluate LLMs with **GSM8K** [73] and **MATH** [74] for math, **HumanEval** [75] and **MBPP** [76] for code capability. We conduct supervised fine-tuning with **MetaMath** [77] and **Code-Feedback** [4] for math and code, respectively, to ensure there is no data leakage.

**iii.** For the multi-modal understanding datasets, we include **GQA** [78], **ScienceQA** (79, **SQA** in Table 4), **TextVQA** (80, **VQA**[T] in Table 4) and **POPE** [81] to test how well the trained model fits with the vision projector and understands the images.

**iv.** We also adopt the GLUE benchmark [5] for the NLU tasks, which consists of the following datasets:

---

[4] https://huggingface.co/datasets/HuggingFaceH4/Code-Feedback
[5] https://huggingface.co/datasets/nyu-mll/glue

a) Single Sentence Classification Tasks: **SST-2** [82] or the Stanford Sentiment Treebank's goal is to predict the sentiment (*positive/negative*) of reviews on different movies, which is a binary classification task. **CoLA** [83] or the Corpus of Linguistic Acceptability consists of sentences each annotated with whether it is a grammatical English sentence.

b) Similarity or Paraphrase Tasks: **MRPC** [84] or the Microsoft Research Paraphrase Corpus is to identify if a sentence pair consists of sentences paraphrases of each other. **QQP** or Quora Question Pairs [6] is to determine whether two questions are semantically equivalent, question pairs are collected from the website Quora. **STS-B** [85] or the Semantic Textual Similarity Benchmark is a collection of sentence pairs drawn from news headlines, video and image captions, and natural language inference data. The task is to evaluate how similar two chunks of texts are with a score from 1 to 5.

c) Language Entailment Tasks: **MNLI** [86] or the Multi-Genre Natural Language Inference is a crowdsourced dataset of sentence pairs with entailment annotations, sourced from diverse materials like speech, fiction, and reports, evaluated on both in-domain and cross-domain sections using private labels. **QNLI** or the Question Natural Language Inference consists of question-paragraph pairs from Wikipedia, originally from the SQuAD [87] and post processed when building GLUE. **RTE** or the Recognizing Textual Entailment is a binary entailment task with a small training dataset, which consists of sentence pairs from four annual textual entailment challenges [88–91].

---

[6]`https://www.kaggle.com/datasets/quora/question-pairs-dataset`

