# OpenReview forum: "RidgeLoRA: Matrix Ridge Enhanced Low-Rank Adaptation of Large Language Models"
_NeurIPS.cc/2025/Conference — NeurIPS 2025 spotlight_

### Official Review · Reviewer_PKfP · 2025-06-26

**Clarity:** 4
**Significance:** 3
**Originality:** 3
**Rating:** 5
**Confidence:** 4

**Summary:**

The paper proposes **RidgeLoRA**, a novel PEFT method that replaces LoRA’s parallel structure with a series connection and adds a diagonal ridge term to enhance expressiveness. Combined with improved initialization and a rank-promoting loss, RidgeLoRA achieves strong performance and efficiency, consistently outperforming prior approaches and offering new insights into PEFT design.

**Questions:**

Can the authors provide further empirical justification for why this change leads to improved representation and learning dynamics? For example, include further analysis or ablation comparing intermediate representations (e.g., gradients) between series and parallel designs to explain the observed performance gains.

**Ethical Concerns:**

["NO or VERY MINOR ethics concerns only"]

**Final Justification:**

I believe this is a very good paper, though it doesn’t quite reach the level of having a groundbreaking impact. Therefore, I will maintain my decision to accept it.

**Limitations:**

Yes

**Paper Formatting Concerns:**

No.

**Quality:**

4

**Strengths And Weaknesses:**

Strengths:
1. The paper provides a rigorous mathematical derivation of RidgeLoRA’s representational advantage over standard LoRA. The inclusion of Theorem 3.1 and detailed analysis underpins the claims with solid theory.

2. The work relaxes the strict initialization constraints of LoRA, enabling broader and more effective initialization strategies.

3. By mitigating the rank deficiency in LoRA through a lightweight ridge term, the paper tackles a core bottleneck in low-rank adaptation and shows improvements across strong baselines.

4. Extensive empirical evaluations across a range of tasks (commonsense reasoning, math/code generation, and multi-modal understanding) using multiple backbone models (LLaMA-2, LLaMA-3.1, Mistral) validate RidgeLoRA’s effectiveness.

5. RidgeLoRA is lightweight, and can be easily implemented in existing LoRA pipelines with minimal code changes. This makes it practical for both research and production use.

Weaknesses:

1. The paper acknowledges the absence of scaling experiments due to resource limitations, which slightly limits conclusions on its applicability to frontier LLMs.

---

> ### Author Rebuttal · Authors · 2025-07-30
>
> We sincerely appreciate Reviewer PKfP's insightful review. Our point-by-point responses are provided as follows.
>
> > The paper acknowledges the absence of scaling experiments due to resource limitations, which slightly limits conclusions on its applicability to frontier LLMs.
>
> We appreciate that reviewers have pointed the scaling experiments out. Indeed, experiments with larger models are absolutely crucial. We have now included additional scaling experiments with Llama-2-13b to address this concern. The results further validate our method’s effectiveness on larger, frontier LLMs.
>
> |                         | **BoolQ** | **PIQA** | **SIQA** | **HellaSwag** | **WinoGrande** | **ARC-e** | **ARC-c** | **OBQA** | **Avg** |
> | ----------------------- | --------- | -------- | -------- | ------------- | -------------- | --------- | --------- | -------- | ------- |
> | Llama-2-13b (LoRA)      | 68.66     | 82.86    | 78.97    | 92.36         | 77.58          | 85.56     | 68.00     | 76.40    | 78.80   |
> | Llama-2-13b (DoRA)      | 70.62     | 84.71    | 78.81    | 93.17         | 79.24          | 88.17     | 74.91     | 77.80    | 80.93   |
> | Llama-2-13b (PiSSA)     | 71.90     | 85.85    | 82.14    | 95.71         | 82.56          | 88.01     | 73.63     | 80.80    | 82.57   |
> | Llama-2-13b (KaSA)      | 69.61     | 83.35    | 79.53    | 92.87         | 78.37          | 86.49     | 69.03     | 76.80    | 79.51   |
> | Llama-2-13b (RidgeLoRA) | 74.20     | 87.76    | 82.50    | 95.22         | 86.42          | 90.28     | 78.07     | 83.60    | 84.76   |
>
> From this table, it can be seen that as the model parameter scale increases, RidgeLoRA consistently outperforms other baselines, with an average improvement of 4% over vanilla LoRA.
>
> > Can the authors provide further empirical justification for why this change leads to improved representation and learning dynamics? For example, include further analysis or ablation comparing intermediate representations (e.g., gradients) between series and parallel designs to explain the observed performance gains.
>
> We thank Reviewer PKfP for this good suggestion that can definitely provide more evidence for the improved training efficiency. We tracked the gradient norm (`grad_norm`) over training steps for RidgeLoRA, PiSSA, KaSA, DoRA and vanilla LoRA. RidgeLoRA and PiSSA showcase higher gradient norms than other LoRA variants, indicating stronger learning dynamics. This supports that RidgeLoRA improves representation and training efficiency. We will add the curve of `grad_norm` from different variants to the revised version.
>
> | **step** | **LoRA** | **DoRA** | **KaSA** | **PiSSA** | **RidgeLoRA** |
> | -------- | -------- | -------- | -------- | --------- | ------------- |
> | 0        | 1.393    | 1.729    | 1.415    | 31.488    | 45.469        |
> | 1        | 1.425    | 1.763    | 1.447    | 32.165    | 46.546        |
> | 2        | 1.429    | 1.761    | 1.446    | 32.018    | 46.193        |
> | 3        | 1.394    | 1.726    | 1.42     | 31.466    | 44.341        |
> | 4        | 1.421    | 1.76     | 1.445    | 31.915    | 43.388        |
> | 5        | 1.424    | 1.762    | 1.447    | 31.504    | 40.92         |
> | 6        | 1.43     | 1.769    | 1.454    | 30.527    | 38.775        |
> | 7        | 1.428    | 1.763    | 1.45     | 30.193    | 36.892        |
> | 8        | 1.422    | 1.753    | 1.441    | 29.606    | 34.801        |
> | 9        | 1.454    | 1.78     | 1.472    | 27.18     | 32.838        |
> | 10       | 1.468    | 1.788    | 1.49     | 26.729    | 32.144        |
> | 11       | 1.489    | 1.809    | 1.507    | 26.221    | 30.791        |
> | 12       | 1.493    | 1.809    | 1.513    | 23.528    | 31.732        |
> | 13       | 1.58     | 1.875    | 1.593    | 23.203    | 31.284        |
> | 14       | 1.598    | 1.889    | 1.616    | 22.453    | 23.761        |
> | 15       | 1.648    | 1.918    | 1.655    | 22.676    | 30.052        |
> | ...      | ...      | ...      | ...      | ...       | ...           |
> | 995      | 0.302    | 0.286    | 0.295    | 1.273     | 1.127         |
> | 996      | 0.422    | 0.405    | 0.409    | 1.102     | 1.425         |
> | 997      | 0.332    | 0.321    | 0.315    | 0.938     | 1.184         |
> | 998      | 0.303    | 0.293    | 0.307    | 0.822     | 0.985         |
> | 999      | 0.273    | 0.279    | 0.273    | 1.524     | 0.943         |
> | 1000     | 0.344    | 0.336    | 0.340    | 1.316     | 1.335         |
> | 1001     | 0.397    | 0.393    | 0.397    | 1.015     | 1.084         |

---

### Official Review · Reviewer_S5wb · 2025-06-27

**Clarity:** 3
**Significance:** 3
**Originality:** 2
**Rating:** 4
**Confidence:** 4

**Summary:**

The authors propose a PEFT method called RidgeLoRA which makes use of an extra trainable diagonal ridge term in addition to the low-rank factorization used in LoRA. They show their method performs better than some of the state-of-art PEFT methods like DoRA and PiSSA. There is also good ablation studies on the importance of the ridge term.

**Questions:**

- Why could RidgeLoRA use fewer parameters than LoRA for Llama-2-7B in Table 2 and Table 3 when it has extra diagonal parameters?

- The use of ridge term also bears some resemblance to the method IA^3, which makes use of multiplicative diagonal scaling of the weight matrices as trainable parameters. The authors can consider including it in the discussion of related works.

Few-Shot Parameter-Efficient Fine-Tuning is Better and Cheaper than In-Context Learning
Liu et al 2022

**Ethical Concerns:**

["NO or VERY MINOR ethics concerns only"]

**Final Justification:**

I am keeping my recommendation of borderline accept/accept since most reviewers agree that this is a solid paper, and the authors have sufficiently addressed our concerns. My minor complaints were over the use of terminologies and writing clarity issues, but I believe the authors would be able to fix them.

**Limitations:**

Yes

**Quality:**

3

**Strengths And Weaknesses:**

- The authors propose an interesting PEFT method called RidgeLoRA, which makes use of multiplicative updates to the weights and an extra ridge term. They show that their method performs better than some of the state-of-art PEFT methods like DoRA and PiSSA. They also have good set of ablation studies on their design choices. Overall it is quite clearly written and a solid paper.

- There is a set of good ablation studies performed to evaluate the different design choices of the method. Of particular importance is Table 6 where the authors show the importance of adding the ridge term, which helps other PEFT methods too.

- For weaknesses there seems to be some mismatch in the comparison with LoRA. The current RidgeLoRA algorithm is a multiplicative update to W while LoRA is an additive update to W. It is not an apples-to-apples to comparison between the rank of a additive and the rank of a multiplicative update. So apart from the trainable ridge term \Sigma, has the authors considered comparing the effect of additive VS multiplicative updates by setting \Sigma = I?

- I also find the use of the terms serial and parallel as illustrated in Figure 1 slightly confusing. It seems to me much more natural to describe the updates as additive or multiplicative to the weight matrix W.

- Theorem 3.1 is related to the approximation property of adding a ridge term to matrices in general, and does not seem to have any direct relevance to finetuning performance. We all know that adding more parameters or increasing the rank should decrease the approximation error, so the theorem doesn't really contain any special insight on why the algorithm works.

---

> ### Author Rebuttal · Authors · 2025-07-30
>
> We sincerely appreciate Reviewer S5wb's insightful comments and suggestions. Our point-by-point responses are provided below.
>
> > The current RidgeLoRA algorithm is a multiplicative update to W while LoRA is an additive update to W. It is not an apples-to-apples to comparison between the rank of a additive and the rank of a multiplicative update. So apart from the trainable ridge term \Sigma, has the authors considered comparing the effect of additive VS multiplicative updates by setting \Sigma = I?
>
> Thank you for this suggestion. To better isolate the effect of the multiplicative structure from the trainable ridge term $\Sigma$, we conducted an additional experiment where we kept the ridge values **frozen** as the identity matrix (i.e., $\Sigma = I$). This effectively removes the contribution of the trainable ridge term while preserving the multiplicative nature of the update.
>
> |                          | **BoolQ** | **PIQA** | **SIQA** | **HellaSwag** | **WinoGrande** | **ARC-e** | **ARC-c** | **OBQA** | **Avg** |
> | ------------------------ | --------- | -------- | -------- | ------------- | -------------- | --------- | --------- | -------- | ------- |
> | Llama-2-7b (Ridge)       | 71.43     | 83.08    | 81.27    | 93.24         | 81.53          | 86.66     | 71.76     | 82.00    | 81.37   |
> | Llama-2-7b (Freeze)      | 70.83     | 82.92    | 80.19    | 93.34         | 80.98          | 86.20     | 71.84     | 81.80    | 81.01   |
> | Llama-3.1-8B (Ridge)     | 73.90     | 89.34    | 83.67    | 95.79         | 86.98          | 93.52     | 83.87     | 87.00    | 86.76   |
> | Llama-3.1-8B (Freeze)    | 73.13     | 89.45    | 83.52    | 95.72         | 87.06          | 93.64     | 81.91     | 87.60    | 86.50   |
> | Mistral-v0.3-7B (Ridge)  | 74.63     | 91.35    | 81.88    | 96.38         | 88.63          | 93.18     | 82.59     | 90.80    | 87.43   |
> | Mistral-v0.3-7B (Freeze) | 73.35     | 91.46    | 83.27    | 96.34         | 88.56          | 92.97     | 81.06     | 90.4     | 87.18   |
>
> We observe that although the performance of this is slightly weaker than the full RidgeLoRA with a learnable $\Sigma$, it still significantly outperforms standard additive LoRA and most of its known variants. This result suggests that the **multiplicative structure itself contributes meaningfully to performance improvements**, even without additional trainable parameters from $\Sigma$. In other words, the benefit of RidgeLoRA is not solely due to parameter count or flexibility, but also due to the structural advantage of multiplicative adaptation.
>
> > I also find the use of the terms serial and parallel as illustrated in Figure 1 slightly confusing. It seems to me much more natural to describe the updates as additive or multiplicative to the weight matrix W.
>
> Thank you for the suggestion. We agree that describing the updates as *additive* or *multiplicative* to the weight matrix W may be more intuitive than using the terms *parallel* and *serial*. In the revision, we will revise the terminology and clarify the expressions—i.e., $X(W + AB)$ (additive) and $X(\lambda \Sigma + AB)W$ (multiplicative)—to improve clarity and readability.
>
> > Theorem 3.1 is related to the approximation property of adding a ridge term to matrices in general, and does not seem to have any direct relevance to finetuning performance. We all know that adding more parameters or increasing the rank should decrease the approximation error, so the theorem doesn't really contain any special insight on why the algorithm works.
>
> We thank Reviewer S5wb for the thoughtful comment. While Theorem 3.1 primarily addresses the approximation error, this quantity is tightly connected to the model's ability to fit the parameter updates during finetuning. In particular, a lower approximation error indicates that the modified weight matrix more faithfully captures the intended update directions. Theorem 3.1 thus provides a theoretical foundation for why adding the ridge term leads to better representational fidelity. Rather than being a generic observation that more parameters help, the theorem quantitatively shows how the ridge term specifically improves the approximation in a non-trivial manner, helping explain the empirical gains we observe.
>
> > Why could RidgeLoRA use fewer parameters than LoRA for Llama-2-7B in Table 2 and Table 3 when it has extra diagonal parameters?
>
> We thank Reviewer S5wb for carefully examining our work, here we further clarify the details of RidgeLoRA to avoid misunderstandings. As illustrated in Figure 1, the reason RidgeLoRA can use fewer parameters than LoRA despite introducing an additional diagonal term lies in the structural difference between the two methods.
>
> While vanilla LoRA adopts a parallel connection (or additive) structure of the form $X(W+AB)$, RidgeLoRA employs a *series connection* (or multiplicative) formalized as $X(\lambda\Sigma + AB)W$. This formulation not only introduces the ridge term $\Sigma$, but also changes the shape of matrix B from $d_{out}\times r$ to $d_{in} \times r$, as reflected in the right part of Figure 1.
>
> Because of this change, the number of trainable parameters in $B$ is reduced. As a result, even with the added diagonal matrix $\Sigma$, which is rather lightweight compared with $B$, RidgeLoRA maintains a lower overall parameter size. Comparisons on the parameter size can also be found in the second column of Table 1.
>
> > The use of ridge term also bears some resemblance to the method IA^3, which makes use of multiplicative diagonal scaling of the weight matrices as trainable parameters. The authors can consider including it in the discussion of related works.
>
> We sincerely thank Reviewer S5wb for pointing this out. IA^3[1] introduces trainable multiplicative diagonal scaling to adapt pre-trained models. As another PEFT method that is different from LoRA and its variants, IA^3 bears conceptual resemblance to RidgeLoRA’s use of the ridge term. We will include a discussion of IA^3 in the related work section and add the appropriate citation in the revised version.
>
> [1] Haokun Liu, Derek Tam, Mohammed Muqeeth, Jay Mohta, Tenghao Huang, Mohit Bansal, and Colin A Raffel.2022. Few-shot parameter-efficient fine-tuning is better and cheaper than in-context learning. Advances in Neural Information Processing Systems 35 (2022), 1950–1965.

---

### Official Review · Reviewer_WvKZ · 2025-07-02

**Clarity:** 3
**Significance:** 3
**Originality:** 3
**Rating:** 5
**Confidence:** 3

**Summary:**

This work proposes a novel low-rank adaptation approach for fine-tuning LLMs. In particular, they propose an approach where the low-rank adapter layers are applied after the linear weights instead of in parallel, and a diagonal matrix is added to each low-rank matrix decomposition to approximate full-rank updates. Additionally, they introduce an additional loss term that encourages the adapter weights to have higher rank, to ensure that high-rank regions of the parameter space are explored throughout training. Experiments were conducted at the 7-8B-parameter scale across a variety of downstream tasks (e.g., commonsense reasoning, math and coding) with comparisons against other low-rank approaches and full fine-tuning.

**Questions:**

Please see my questions in the Weaknesses section.

**Ethical Concerns:**

["NO or VERY MINOR ethics concerns only"]

**Final Justification:**

All of my remaining concerns have been addressed during the discussion period. I keep my positive view of the paper and maintain the original score of 5.

**Limitations:**

Yes.

**Quality:**

3

**Strengths And Weaknesses:**

Strengths:
- Presents a novel approach that improves the representational capacity of low-rank adaptation while maintaining computational efficiency.
- Evaluations are comprehensive, covering a variety of downstream tasks and ablation studies, and demonstrate that the proposed RidgeLoRA approach can result in performance improvements over existing low-rank approaches and occasionally performance on par with that of full fine-tuning.

Weaknesses:
- Evaluations are limited to the 7-8B model scale, and it is possible that the approximation quality and thus the gap vs. the full fine-tuning baseline shows different trends at other model scales.
- Performance improvements are relatively small vs. other low-rank baselines with more recent LLMs (Llama-3.1-8B and Mistral-v0.3-7B).
- The nuclear norm plot in Figure 2 appears to suggest that the weight updates still remain relatively low rank (similar to that of PiSSA) on GQA. How do these trends change across different datasets? On a related note, prior works suggest that the required rank r on the adapters can vary significantly across different downstream tasks (i.e., they have different intrinsic dimensionalities) [1]. Perhaps an interesting study would be to compare the same set of methods across tasks representative of different intrinsic dimensionalities to demonstrate that the proposed method is indeed more effective at approximating more challenging tasks that require higher rank updates.
- It is not completely obvious to me whether the series connection necessarily has an advantage over the parallel approach in terms of predictive performance. The improvements observed in Table 6 seem very marginal. Perhaps the claim about the necessity of the serial connection requires additional validation.
- Could the authors provide additional explanation as to why the numbers for Llama-3.1-8B and Mistral-v0.3-7B in Table 2 are often much worse than all of the low-rank methods? I am aware that the paper mentions this is similar to the trends observed in other papers, but am wondering if adding sufficient regularization can update the numbers to be more reflective of true full fine-tuning performance.

References:
[1] LoRA Learns Less and Forgets Less (Biderman et al., 2024)

---

> ### Author Rebuttal · Authors · 2025-07-30
>
> We appreciate the review of Reviewer WvKZ for its constructive suggestions. Our detailed rebuttal is as follows.
>
> > The nuclear norm plot in Figure 2 appears to suggest that the weight updates still remain relatively low rank (similar to that of PiSSA) on GQA.
>
> Analyzing rank behavior across tasks with varying intrinsic dimensionality is indeed insightful. In response, we performed additional analysis using checkpoints from a diverse set of tasks (model weights trained on MetaMathQA, CommonSense, and Code-Feedback) and found that RidgeLoRA consistently yields higher nuclear norms compared to other LoRA variants. This suggests that RidgeLoRA is more capable of adapting to tasks that require higher-rank updates.
>
> Due to the constraints of the rebuttal format (e.g., no figures or external links), we are unable to include the full visualizations and detailed comparisons here. However, we will provide comprehensive nuclear norm plots and cross-task analysis in the final version of the paper.
>
> > It is not completely obvious to me whether the series connection necessarily has an advantage over the parallel approach in terms of predictive performance. The improvements observed in Table 6 seem very marginal. Perhaps the claim about the necessity of the serial connection requires additional validation.
>
> Here we include results as instructed by Reviewer S5wb to prove that the multiplicative structure (or serial connection) itself contributes a lot to the good performance.
>
> |                          | **BoolQ** | **PIQA** | **SIQA** | **HellaSwag** | **WinoGrande** | **ARC-e** | **ARC-c** | **OBQA** | **Avg** |
> | ------------------------ | --------- | -------- | -------- | ------------- | -------------- | --------- | --------- | -------- | ------- |
> | Llama-2-7b (Ridge)       | 71.43     | 83.08    | 81.27    | 93.24         | 81.53          | 86.66     | 71.76     | 82.00    | 81.37   |
> | Llama-2-7b (Freeze)      | 70.83     | 82.92    | 80.19    | 93.34         | 80.98          | 86.20     | 71.84     | 81.80    | 81.01   |
> | Llama-3.1-8B (Ridge)     | 73.90     | 89.34    | 83.67    | 95.79         | 86.98          | 93.52     | 83.87     | 87.00    | 86.76   |
> | Llama-3.1-8B (Freeze)    | 73.13     | 89.45    | 83.52    | 95.72         | 87.06          | 93.64     | 81.91     | 87.60    | 86.50   |
> | Mistral-v0.3-7B (Ridge)  | 74.63     | 91.35    | 81.88    | 96.38         | 88.63          | 93.18     | 82.59     | 90.80    | 87.43   |
> | Mistral-v0.3-7B (Freeze) | 73.35     | 91.46    | 83.27    | 96.34         | 88.56          | 92.97     | 81.06     | 90.4     | 87.18   |
>
> To better isolate the effect of the multiplicative structure from the trainable ridge term $\Sigma$, we conducted an additional experiment where we kept the ridge values **frozen** as the identity matrix (i.e., $\Sigma = I$). This effectively removes the contribution of the trainable ridge term while preserving the multiplicative nature of the update.
>
> We observe that although the performance of this is slightly weaker than the full RidgeLoRA with a learnable $\Sigma$, it still significantly outperforms standard additive LoRA and most of its known variants. This result suggests that the **multiplicative structure itself contributes meaningfully to performance improvements**, even without additional trainable parameters from $\Sigma$. In other words, the benefit of RidgeLoRA is not solely due to parameter count or flexibility, but also due to the structural advantage of multiplicative adaptation.
>
> > Could the authors provide additional explanation as to why the numbers for Llama-3.1-8B and Mistral-v0.3-7B in Table 2 are often much worse than all of the low-rank methods? I am aware that the paper mentions this is similar to the trends observed in other papers, but am wondering if adding sufficient regularization can update the numbers to be more reflective of true full fine-tuning performance.
>
> As discussed in the paper, similar trends have been reported in prior work (e.g., DoRA, PiSSA, and KaSA), particularly in instruction tuning settings, where full fine-tuning (FFT) can be prone to overfitting or instability without careful regularization. In this context, LoRA can be seen as a form of regularization that constrains the parameter update space, which may help explain why it sometimes outperforms unconstrained FFT. Our work focuses specifically on the low-rank setups, and comprehensive results showcase that RidgeLoRA outperforms other LoRA variants.
>
> In our experiments, we adopted commonly used FFT configurations without extensive hyper-parameter tuning. One key reason for this choice is fairness: although our primary focus is on comparing **low-rank adaptation methods**, we include FFT (training without regularization on weight updates) as a reference. To ensure meaningful comparisons, all methods within the same category (FFT or low-rank) are trained under consistent hyper-parameters. That is, all low-rank methods, including RidgeLoRA, share the same setup, and all FFT experiments follow a fixed default configuration.

---

> > ### Comment · Reviewer_WvKZ · 2025-08-08
> > **Response by Reviewer**
> >
> > I thank the authors for providing a response to my questions and concerns. My other concerns noted in W4 and W5 have been sufficiently addressed. Regarding W3, please see below.
> >
> > > In response, we performed additional analysis using checkpoints from a diverse set of tasks (model weights trained on MetaMathQA, CommonSense, and Code-Feedback) and found that RidgeLoRA consistently yields higher nuclear norms compared to other LoRA variants. This suggests that RidgeLoRA is more capable of adapting to tasks that require higher-rank updates. Due to the constraints of the rebuttal format (e.g., no figures or external links), we are unable to include the full visualizations and detailed comparisons here. However, we will provide comprehensive nuclear norm plots and cross-task analysis in the final version of the paper.
> >
> > I understand that there are constraints with the rebuttal format, but would it be possible to share some of the nuclear norm numbers (and the trends) that you are referring to? To be clear, I am not expecting that the authors provide a full-on, unreasonably comprehensive set of results here, but at least some discussion of quantitative results would be helpful here. I am assuming that you already have some of these additional results, based on the author's comment.

---

> ### Author Response · Authors · 2025-08-09
> **Response by Authors**
>
> We greatly appreciate Reviewer WvKZ’s feedback on our rebuttal and are pleased to share the detailed nuclear norm results of weight updates across different datasets. Figure 2 from the original submission is on CommonSense dataset, while the others are presented as follows:
>
> | Task          | Method    | Nuclear Norm of $W_Q$ | Nuclear Norm of $W_K$ | Nuclear Norm of $W_V$ | Nuclear Norm of $W_O$ |
> | ------------- | --------- | ---------- | ---------- | ---------- | ---------- |
> |MetaMathQA| FFT | 105.2 | 39.7 | 44.0 | 98.3 |
> || RidgeLoRA | 80.4 | 28.5 | 27.4 | 75.1 |
> || PiSSA     | 61.6 | 20.9 | 19.8 | 58.2 |
> || KaSA      | 53.2 | 18.6 | 16.9 | 50.8 |
> || LoRA      | 44.1 | 11.6 | 9.3 | 45.5 |
> |Code-Feedback| FFT | 131.7 | 45.9 | 49.3 | 125.8 |
> || RidgeLoRA | 95.4 | 32.9 | 30.7 | 83.9 |
> || PiSSA     | 81.3 | 29.7 | 28.5 | 82.0 |
> || KaSA      | 69.4 | 23.3 | 18.2 | 71.9 |
> || LoRA      | 52.7 | 18.4 | 15.2 | 49.3 |
>
> From these results, we observe that RidgeLoRA consistently achieves significantly higher nuclear norms than other LoRA variants (PiSSA, KaSA, vanilla LoRA), indicating a greater capacity to induce higher-rank updates. This trend holds robustly across multiple tasks, supporting our claim that RidgeLoRA is better suited for tasks requiring richer adaptation compared to standard LoRA methods. We will also properly discuss our observations together with those from [1] in the revised version.
>
> Also, for concern W1 and W2, as instructed by other reviewers, here we provide results from Llama-2-13B to address  concerns w.r.t. other model scales.
>
> > Evaluations are limited to the 7-8B model scale, and it is possible that the approximation quality and thus the gap vs. the full fine-tuning baseline shows different trends at other model scales.
>
>
>
> |                         | **BoolQ** | **PIQA** | **SIQA** | **HellaSwag** | **WinoGrande** | **ARC-e** | **ARC-c** | **OBQA** | **Avg** |
> | ----------------------- | --------- | -------- | -------- | ------------- | -------------- | --------- | --------- | -------- | ------- |
> | Llama-2-13b (LoRA)      | 68.66     | 82.86    | 78.97    | 92.36         | 77.58          | 85.56     | 68.00     | 76.40    | 78.80   |
> | Llama-2-13b (DoRA)      | 70.62     | 84.71    | 78.81    | 93.17         | 79.24          | 88.17     | 74.91     | 77.80    | 80.93   |
> | Llama-2-13b (PiSSA)     | 71.90     | 85.85    | 82.14    | 95.71         | 82.56          | 88.01     | 73.63     | 80.80    | 82.57   |
> | Llama-2-13b (KaSA)      | 69.61     | 83.35    | 79.53    | 92.87         | 78.37          | 86.49     | 69.03     | 76.80    | 79.51   |
> | Llama-2-13b (RidgeLoRA) | 74.20     | 87.76    | 82.50    | 95.22         | 86.42          | 90.28     | 78.07     | 83.60    | 84.76   |
>
> > Performance improvements are relatively small vs. other low-rank baselines with more recent LLMs (Llama-3.1-8B and Mistral-v0.3-7B).
>
> As noted, the relative gains on newer LLMs (Llama-3.1-8B, Mistral-v0.3-7B) are smaller compared to older models. This is expected, as more recent base models are trained with improved synthesized data and carefully designed instruction-tuning recipes, leaving less headroom for further improvement by alternative low-rank variants.
>
> Nevertheless, RidgeLoRA still delivers consistent gains in these settings without introducing additional parameters or computational overhead. These improvements remain practically meaningful and can be readily applied to other domains that require low-rank adaptations.
>
> [1] Biderman et al., LoRA Learns Less and Forgets Less, Transactions on Machine Learning Research, 2024.

---

> > ### Comment · Reviewer_WvKZ · 2025-08-09
> > **Response by Reviewer**
> >
> > Thank you for the follow-up and the detailed responses, I really appreciate it. I keep my positive view of the paper and maintain my score of 5.

---

### Official Review · Reviewer_pZr9 · 2025-07-05

**Clarity:** 3
**Significance:** 3
**Originality:** 3
**Rating:** 5
**Confidence:** 3

**Summary:**

This paper presents RidgeLoRA, a PEFT method for LLMs that addresses the representation limitations of standard LoRA. By incorporating a full-rank ridge module through a diagonal matrix and redesigning the architecture from parallel to series connections, RidgeLoRA enables more accurate approximation of full-rank weight updates while maintaining efficiency. Experiments across multiple domains demonstrate that RidgeLoRA outperforms other LoRA variants and often matches or exceeds the performance of full-rank training.

**Questions:**

1. Could the authors provide a more detailed analysis of RidgeLoRA’s performance on models larger than 8B parameters?
2. Could the authors provide case studies to illustrate RidgeLoRA’s superior performance?

**Ethical Concerns:**

["NO or VERY MINOR ethics concerns only"]

**Final Justification:**

Thanks for your response. The rebuttal has addressed my concerns. I will keep my positive score.

**Limitations:**

yes

**Quality:**

3

**Strengths And Weaknesses:**

Strength:

1. Extensive experiments are conducted across multiple domains and a range of large language models.
2. The writing is clear and the paper is easy to follow.

Weakness:

1. The evaluation is limited to models with up to 8B parameters; the performance of RidgeLoRA on larger models (e.g., 70B) remains unclear.

---

> ### Author Rebuttal · Authors · 2025-07-30
>
> We thank Reviewer pZr9 for the professional review. Our rebuttal is as follows.
>
> > Could the authors provide a more detailed analysis of RidgeLoRA’s performance on models larger than 8B parameters?
>
> We appreciate that reviewers have pointed the scaling experiments out. Indeed, experiments with larger models are absolutely crucial. We have now included additional scaling experiments with Llama-2-13b to address this concern. The results further validate our method’s effectiveness on larger, frontier LLMs.
>
> |                         | **BoolQ** | **PIQA** | **SIQA** | **HellaSwag** | **WinoGrande** | **ARC-e** | **ARC-c** | **OBQA** | **Avg** |
> | ----------------------- | --------- | -------- | -------- | ------------- | -------------- | --------- | --------- | -------- | ------- |
> | Llama-2-13b (LoRA)      | 68.66     | 82.86    | 78.97    | 92.36         | 77.58          | 85.56     | 68.00     | 76.40    | 78.80   |
> | Llama-2-13b (DoRA)      | 70.62     | 84.71    | 78.81    | 93.17         | 79.24          | 88.17     | 74.91     | 77.80    | 80.93   |
> | Llama-2-13b (PiSSA)     | 71.90     | 85.85    | 82.14    | 95.71         | 82.56          | 88.01     | 73.63     | 80.80    | 82.57   |
> | Llama-2-13b (KaSA)      | 69.61     | 83.35    | 79.53    | 92.87         | 78.37          | 86.49     | 69.03     | 76.80    | 79.51   |
> | Llama-2-13b (RidgeLoRA) | 74.20     | 87.76    | 82.50    | 95.22         | 86.42          | 90.28     | 78.07     | 83.60    | 84.76   |
>
> From this table, it can be seen that as the model parameter scale increases, RidgeLoRA consistently outperforms other baselines, with an average improvement of 4% over vanilla LoRA.
>
> > Could the authors provide case studies to illustrate RidgeLoRA’s superior performance?
>
> Thank you for the suggestion. Here we provide an illustrative case demonstrating RidgeLoRA’s superior reasoning capability compared to vanilla LoRA tuning.
>
> Consider the following question:
>
> ```
> Question: A rooster lays an egg on the top of a slanted roof. Which way does the egg roll?
> A. To the left, because the roof is slanted that way.
> B. To the right, since the slope leads downward.
> C. It stays on the roof because it's balanced.
> D. It doesn’t roll at all—roosters don’t lay eggs.
> ```
>
> The Llama model fine-tuned with vanilla LoRA answers:
>
> ```
> The answer is B. To the right, since the slope leads downward.
> ```
>
> In contrast, the RidgeLoRA-enhanced Llama model correctly identifies the trick and answers:
>
> ```
> The answer is D. It doesn’t roll at all—roosters don’t lay eggs. Roosters don’t lay eggs, so this is a trick question.
> ```
>
> This example shows that RidgeLoRA can better capture nuanced reasoning and common-sense knowledge. We will include more case studies in other domains such as math and code problems in the revised version to further illustrate RidgeLoRA’s advantages.

---

> > ### Comment · Reviewer_pZr9 · 2025-08-07
> >
> > Thanks for your response. The rebuttal has addressed my concerns. I will keep my positive score.

---

### Decision · Program_Chairs · 2025-09-17

**Decision:**

Accept (spotlight)

**Comment:**

This paper proposes RidgeLoRA, a parameter-efficient fine-tuning method for language models. RidgeLoRA introduces a diagonal component alongside LoRA weights and enhances their representation capacity through multiplicative updates. Both empirical and theoretical studies demonstrate its advantages over other PEFT methods. All reviewers recognize the paper’s contributions, and the authors have addressed most of their concerns in the rebuttal. I concur with the reviewers and recommend acceptance of the paper.

The authors are encouraged to incorporate the reviewers’ suggestions in the camera-ready version, such as adding the 13B experiment from the rebuttal and improving the clarity of certain terminology as noted by Reviewer S5wb.